# An Update of Lectins from Marine Organisms: Characterization, Extraction Methodology, and Potential Biofunctional Applications

**DOI:** 10.3390/md20070430

**Published:** 2022-06-29

**Authors:** Mirja Kaizer Ahmmed, Shuva Bhowmik, Stephen G. Giteru, Md. Nazmul Hasan Zilani, Parise Adadi, Shikder Saiful Islam, Osman N. Kanwugu, Monjurul Haq, Fatema Ahmmed, Charlene Cheuk Wing Ng, Yau Sang Chan, Md. Asadujjaman, Gabriel Hoi Huen Chan, Ryno Naude, Alaa El-Din Ahmed Bekhit, Tzi Bun Ng, Jack Ho Wong

**Affiliations:** 1Department of Food Sciences, University of Otago, P.O. Box 56, Dunedin 9054, New Zealand or kaizer@cvasu.ac.bd (M.K.A.); stephen.giteru@postgrad.otago.ac.nz (S.G.G.); parise.adadi@postgrad.otago.ac.nz (P.A.); 2Department of Fishing and Post-Harvest Technology, Faculty of Fisheries, Chittagong Veterinary and Animal Sciences University, Chittagong 4225, Bangladesh; 3Centre for Bioengineering and Nanomedicine, Faculty of Dentistry, Division of Health Sciences, University of Otago, P.O. Box 56, Dunedin 9054, New Zealand; shuva.bhowmik@postgrad.otago.ac.nz; 4Department of Fisheries and Marine Science, Noakhali Science and Technology University, Noakhali 3814, Bangladesh; 5Alliance Group Limited, Invercargill 9840, New Zealand; 6Department of Pharmacy, Jashore University of Science and Technology, Jashore 7408, Bangladesh; mnhzilani09@gmail.com; 7Institute for Marine and Antarctic Studies, University of Tasmania, Launceston 7250, Australia; shikdersaiful.islam@gmail.com; 8Fisheries and Marine Resource Technology Discipline, Life Science School, Khulna University, Khulna 9208, Bangladesh; 9Institute of Chemical Engineering, Ural Federal University, Mira Street 28, 620002 Yekaterinburg, Russia; nabayire@gmail.com; 10Department of Fisheries and Marine Bioscience, Jashore University of Science and Technology, Jashore 7408, Bangladesh; mr.haq@just.edu.bd; 11Department of Chemistry, University of Otago, P.O. Box 56, Dunedin 9054, New Zealand; ahmfa773@student.otago.ac.nz; 12Medway Maritime Hospital, Medway NHS Foundation Trust, Kent ME7 5NY, UK; charlene.cw.ng@gmail.com; 13Department of Obstetrics & Gynaecology, LKS Faculty of Medicine, The University of Hong Kong, Hong Kong, China; bomberharo@gmail.com; 14Department of Aquaculture, Faculty of Fisheries and Ocean Sciences, Khulna Agricultural University, Khulna 9100, Bangladesh; manikdof@yahoo.com; 15Division of Science, Engineering and Health Studies, College of Professional and Continuing Education, The Hong Kong Polytechnic University, Hong Kong, China; gabriel.chan@cpce-polyu.edu.hk; 16Department of Biochemistry and Microbiology, Nelson Mandela University, Port Elizabeth 6031, South Africa; ryno.naude@mandela.ac.za; 17School of Life Sciences, The Chinese University of Hong Kong, Hong Kong, China; tzibunng@cuhk.edu.hk; 18School of Health Sciences, Caritas Institute of Higher Education, Hong Kong, China

**Keywords:** adhesins, hemagglutinins, marine lectins, characterization, extraction and purification, bio-functional roles, immunomodulation

## Abstract

Lectins are a unique group of nonimmune carbohydrate-binding proteins or glycoproteins that exhibit specific and reversible carbohydrate-binding activity in a non-catalytic manner. Lectins have diverse sources and are classified according to their origins, such as plant lectins, animal lectins, and fish lectins. Marine organisms including fish, crustaceans, and mollusks produce a myriad of lectins, including rhamnose binding lectins (RBL), fucose-binding lectins (FTL), mannose-binding lectin, galectins, galactose binding lectins, and C-type lectins. The widely used method of extracting lectins from marine samples is a simple two-step process employing a polar salt solution and purification by column chromatography. Lectins exert several immunomodulatory functions, including pathogen recognition, inflammatory reactions, participating in various hemocyte functions (e.g., agglutination), phagocytic reactions, among others. Lectins can also control cell proliferation, protein folding, RNA splicing, and trafficking of molecules. Due to their reported biological and pharmaceutical activities, lectins have attracted the attention of scientists and industries (i.e., food, biomedical, and pharmaceutical industries). Therefore, this review aims to update current information on lectins from marine organisms, their characterization, extraction, and biofunctionalities.

## 1. Introduction

The word lectin is derived from the Latin word “Lect, from the verb Legere,” which means “chosen”. According to the definition of the Oxford dictionary, “lectins are any of a class of proteins, chiefly of plant origin, which bind specifically to certain sugars and so cause agglutination of particular cell types”. The sugars or carbohydrate moieties to which lectins have binding potentiality comprise glycoconjugates, glycolipids, monosaccharides, oligosaccharides, and proteoglycans [1,2]. Lectins are ubiquitous and found in prokaryotes and eukaryotes encompassing plants, animals (vertebrates as well as invertebrates), and microorganisms (i.e., bacteria, fungi, protozoa, and viruses) [1,3]. Naturally, lectins exist as monomers, and homo- and hetero-aggregates (i.e., dimers and tetramers). Nonetheless, they are primarily composed of multiple subunits, each with one or multiple carbohydrate recognition domains (CRDs) [1,4]. Presently, there is no universally accepted classification of lectins; hence, they are mostly grouped according to their source (i.e., animal, plant, and microbial lectins). 

In addition, based on the species of origin, lectins are grouped into animal lectins, legume lectins, and grain lectins, among others. Moreover, unlike antibodies which are similar in structure, lectins differ in primary structure, tertiary/quaternary conformation, and molecular weight. Thus, lectins represent a highly diverse group of proteins with regard to their structural aspects and functional roles in various biological processes, which include cell–cell adhesion, cell communication, immune defense, and signal transduction [3,4]. In addition, lectins are classified based on their amino acid sequences, structural homology, and cation requirements. For instance, lectins from marine organisms are generally classified as C-type lectins (CTLs), galectins (formerly called S-type lectins), F-type lectins (FTLs), X-type lectins (XTLs), I-type lectins (ITL), P-type lectins, pentraxins, ricin-type, lily-type lectins, and mannan-binding lectins [5,6]. They are further categorized according to their sugar-binding preferences, into rhamnose-binding lectins (RBL), fucose-binding lectins (FBL), galactose-binding lectins (GBL), GlcNAc-specific lectins, mannan-binding lectins (MBL), and GalNAc-specific lectins. 

Lectins elicit various biological activities at the cellular and molecular levels, which include antitumor, antifungal, antibacterial, and antiviral activities, as well as other possible medicinal applications, and thus could be used to design and develop potent therapeutic agents [1,7], as discussed in Section 5. However, available studies on marine organism- derived lectins have mainly focused on the availability, types, and effects on immunity, i.e., host/organism health [8,9,10,11]. To date, there are no reviews with regards to the bio-functional roles of marine animal lectins from a human health perspective. Therefore, the current review seeks to update the properties, characterization, extraction methodology, and bio-functional roles of lectins derived from marine organisms including fish, crustaceans, and molluscs.

## 2. Marine Organism-Derived Lectins

Over the years, several studies on aquatic animal lectins have largely been motivated by the intent to unravel the functions of these proteins in the innate immune system of the host organism. These vertebrates and invertebrates lack specialized lymphatic organs and rely on pattern-based recognition of “self” and “non-self” cells, as an imperative defense system. Aquatic animals are endowed with a large and complex repertory of lectins. For instance, teleosts and elasmobranchs fish possess lectins in eggs, embryos, mucus, plasma, serum, skin, and other tissues, which confer immunity against infectious diseases [5,12,13]. The biological significance of the isolated lectins in various marine organisms has not been fully elucidated, and investigations are currently ongoing [14]. Nevertheless, their roles in aquatic animals (i.e., reproduction, defense against pathogenic microbes (immunity), polyspermy block, morphogenesis and embryogenesis) are well studied [15,16,17]. The major classes of lectins identified in marine animals are C-type lectins (CTLs), F-type lectins (FTLs), X-type lectins (XTL), galectins (formerly S-type lectins), and pentraxins (P-type lectins) [6,18]. Typically, all lectins mentioned above present similar CRD sequences, the position of disulfide bonds, and calcium ion (Ca^2+^) binding sites. These lectins are collectively involved in the activation of prophenoloxidase (proPO) [19,20], antipathogenic activity, cell agglutination, and other biological responses (Table 1). Typical crystal structures of selected fish lectins are shown in Figure 1 and the details of the major classes are discussed below:

**Figure 1 marinedrugs-20-00430-f001:**
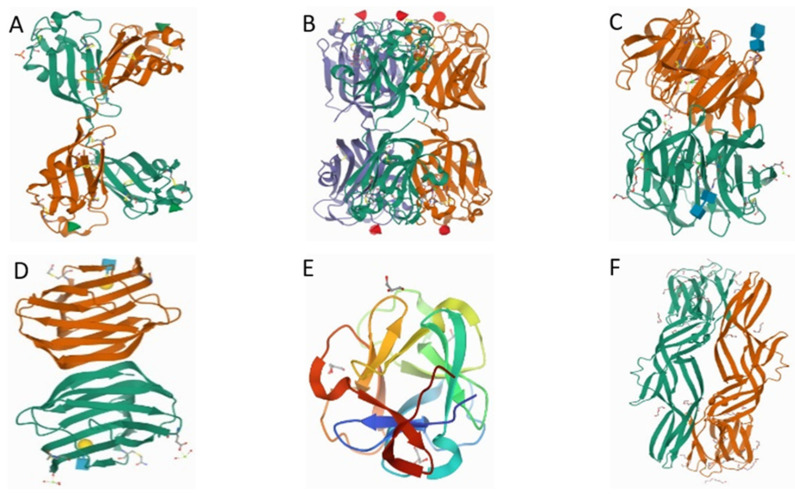
Typical crystal structures of (**A**) rhamnose-binding lectin CLS3 from chum salmon (*Oncorhynchus keta*) [21], (**B**) F-lectin (fucolectin) from striped bass (*Morone saxatilis*) serum [22], (**C**) carp fishelectin (C-type lectin) [23], (**D**) zebrafish *(Dario rerio*) galectin-1-L2 [24], (**E**) *Mytilus californianus* galactose-binding lectin [25], (**F**) zebrafish (*D. rerio*) Dln1 (mannose-binding lectin) [26]. Individual images were obtained from Protein Data Bank (PDB, https://www.rcsb.org/, accessed on 20 June 2022).

### 2.1. Rhamnose Binding Lectins (RBLs)

Rhamnose-binding lectins (RBLs) are lectins mainly associated with fish and invertebrates and recognized by their binding affinity to l-rhamnose [65]. Fish eggs RBLs belong to the sea urchin egg lectin (SUEL)-like lectins, identified by the possession of two or three tandemly repeated CRD domains (d-galactoside/l-rhamnose-binding SUEL) containing 95 amino acid residues [66]. The crystal structure of RBL (CSL3) shows a homodimer of two 20 kDa subunits with N- and C-terminal domains of different subunits forming lobe structures connected with a flexible linker peptide (Figure 1A). The CSL3 evokes apoptosis using globotriaosylceramide (Gb3) as a cellular receptor [21]. RBLs are located mainly in oocytes and ovaries, and skin mucus of fish, which have now been identified in more than 20 fish species [67]. They exhibit unique sequence motifs and structural folds, which are mainly involved in the modulation of fertilization and innate immunity [55]. RBL isolated from the sea bass (*Dicentrarchus labrax*) exhibits a molecular weight between 24 kDa and 100 kDa under reducing and non-reducing conditions, which suggests a homotetrameric structural conformation. The structure comprises 212 amino acid (aa) residues with two tandemly arrayed CRDs and an 18-residue signal sequence at the N-terminal [55]. Contrary to other lectins (i.e., CTL), RBLs do not require Ca^2+^ ions for ligand binding. They act as opsonins that facilitate agglutination of bacteria (typically Gram-negative) and phagocytotic activity of peritoneal macrophages against pathogens (i.e., *E. coli*, among others). In summary, RBLs are involved in the detection and phagocytic elimination of microbial pathogens in blood by binding the pathogens to the leukocyte surface [6]. Multiple l-rhamnose-binding isolectins with various molecular and configurational features have previously been reported in fish eggs and serum. For instance, two l-RBLs, namely STL1 and STL2 extracted from steelhead trout (*Oncorhynchus mykiss*) eggs exhibit distinct molecular weights of 84 and 68 kDa, respectively [56].

Interestingly, both STL1 and STL2 are non-covalently linked trimers made up of 31.4-kDa and 21.5-kDa subunits, respectively. The configuration has been reported to influence their functionalities, such as hemagglutination activity and unique binding specificities to saccharides. Whereas STL1 had a greater affinity for l-arabinose, STL2 exhibited a higher affinity for melibiose. However, the N-acetyl residue inactivated both STL1 and STL2 [56]. The Gb3 carbohydrate chain binding activity of l-RBL isolated from catfish (*Silurus asotus*) eggs has likewise been reported [33,34,68]. *Silurus asotus* lectin (SAL), an α-galactoside-binding protein that is composed of three tandem-repeated domains induced early apoptosis in the Raji cell line (Burk’tt’s lymphoma cell line expressing globotriose, Gb3) without manifesting cytotoxicity. In addition, SAL has been demonstrated as a potent enhancer of the effectiveness of anti-cancer drugs and thus may have a potential for application in cancer therapy [33,34]. Similar homology and Gb3 binding ability were reported in shishamo smelt (*Osmerus lanceolatus*) eggs lectin [69]. Further analysis showed that OLL expressed two tandem-repeated CRD domains which were ten times more reactive in binding and shrinking the Raji cells than SAL. Thus, OLL decreased the growth and viability of cancerous cells more than SAL, which may partly be ascribed to the differences in cell binding properties to the carbohydrate receptor-bearing β-galactoside chains [34].

### 2.2. Fucose-Binding Lectins (FTLs)

Fucose-binding lectins (FTLs), also known as fucolectins, are characterized by a unique amino acid sequence motif, structural fold, and a nominal specificity for l-fucose, a feature of the eel carbohydrate-recognition sequence motif [70]. l-fucose is a non-reducing terminal sugar found in pro-and eukaryotic glycans, which may be released into the human intestinal lumen with the aid of hydrolytic activity of indigenous microbes and pathogens. The crystal structure of fucose-specific lectin from striped bass (*Morone saxatilis*) (Figure 1B) is characterized by two F-type CRDs in tandem, with a cylindrical 81-A-long and 60-A-wide trimer divided into two globular halves. N-terminal CRDs (N-CRDs) and C-terminal CRDs (C-CRDs) can be seen on each half. FTLs extracted from sea bass (*Dicentrarchus labrax* (L.)) [51,52,55] and rock bream (*Oplegnathus fasciatus*) [53] are commonly recognized by their two tandemly arranged CRD with an F-type motif [57]. A 30-kDa Ca^2+^-independent fucose-binding protein with bacterium agglutinating activity has been isolated using affinity chromatography from the serum of the emerald rockcod (*Trematomus bernacchii*) [71].

Furthermore, the fucose-binding lectins (FBLs) isolated from the eggs and embryos of European sea bass (*D. labrax*) expressed characteristics, (i.e., biochemical, immunochemical, and agglutination properties) similar to the previously extracted FBLs [52]. Moreover, they exhibited the ability to trigger agglutination and the destruction of pathogens [51].

Extraction of lectins under reducing and non-reducing conditions yielded proteins with different molecular weights, 34 kDa and 30 kDa, respectively, which may be ascribed to the shrinkage effect induced by the interchain disulfide bridges [52]. FBLs exert potent antibacterial activity and trigger immune defense in response to pathogens in the intestinal mucus and blood of fish [6,51]. Additionally, they can express phagocytic activity against *E. coli*, for example, and the FBLs isolated from hepatocytes and intestinal cells of European eel (*Anguilla anguilla*) and striped bass (*Morone saxatilis*), respectively [57].

### 2.3. C-Type Lectins

C-type lectins (CTLs) are a large group of extracellular proteins composed of at least one carbohydrate-recognition or C-type lectin-like domain with diverse functions. A three-dimensional structure of carp fish-egg lectin and of its complex with N-acetylglucosamine determined by X-ray crystallography (resolutions of 1.35 and 1.70 Å) described previously [23], showed molecule folds as a six-bladed β-propeller and internal short consensus amino-acid sequences (Figure 1C). The structure also depicted a Ca^2+^ ions binding at the bottom of the funnel-shaped tunnel at the center of the propeller. Several CTLs have previously been isolated from the eggs [68], intestine, gills [72], serum, and skin mucus [73,74] of fish. Typically, CTLs contain a double loop, where two highly conserved disulphide bridges, hydrophobic and polar interactions, occur. In addition, one of the loops carries two Ca^2+^-binding domains, which also act as carbohydrate-binding sites [72,75,76]. However, not all CTLs are Ca^2+^-dependent [77]. Surprisingly, CTL receptors are expressed in multiple tissues of the host [78], with potential roles in innate and adaptative immune responses, executed by primarily recognizing invading pathogens [79]. For example, a recent study showed that Atlantic salmon (*Salmo sala*) expressed CTL receptors upon infection by viral and bacterial pathogens [78]. Furthermore, zebrafish (*Danio rerio*) (zhl) liver CTL [80], and zebrafish hepatic lectin-like (zhl-l) CTL [81] can agglutinate and bind to both Gram-negative and Gram-positive bacteria, which enhances phagocytosis. This is owing to the ability of the recombinant C-type lectin domain (rCTLD) of zhl and zhl-l to specifically recognize and bind to galactose and GlcNAc of insoluble lipopolysaccharide (LPS), lipoteichoic acid (LTA), and peptidoglycan (PGN) residues of pathogenic organisms [82]. A recent study [83] identified six CTLs from hepatopancreas and gills of the Manilla clam (*Ruditapes philippinarum*), which depicted a highly conserved amino acid sequence compared to other invertebrate CTL sequences. The CTLs possessed two Ca^2+^ dependent carbohydrate-binding motifs and six-cysteine residues. Four conserved residues were involved in the formation of internal bonds in CRD. A CTL identified in the hemolymph of the giant prawn (*Macrobrachium rosenbergii*) showed Ca^2+^-dependent hemagglutination activity with rabbit red blood cells [84]. The activity was related to the specific binding to N-acetylneuraminic acid and a glycoprotein, fetuin, at minimum inhibitory concentrations of 50 mM and 0.31 mg/mL, respectively.

A C-type lectin-like domain-containing protein (CTLDcps) gene isolated from milkfish (*Chanos chanos*) was upregulated in response to pathogen invasion, emphasizing the role of CTLs in immune defense [85]. The affinity of CTLDcps to sugars, especially galactose, is facilitated by a high density of proline residues [85]. A CTL from weather loach (*Misgurnus anguillicaudatus*) (MaCTL) is composed of 165 amino acids with a C-type lectin domain (CTLD) and a galactose-specific QPD (Gln-Pro-Asp) motif was upregulated by weather loach invasion in a Ca^2+^-dependent manner [86]. Likewise, the expression of Nattectin-like (CaNTC) protein in goldfish (*Carassius auratus*) was upregulated in the liver, spleen, and kidney by *Aeromonas hydrophilia* invasion. It agglutinated bacteria (Gram-negative *E. coli* and *A. hydrophila*, and Gram-positive *Staphylococcus aureus*) in a Ca^2+^-dependent manner, indicating that CaNTC triggered an innate immune response in goldfish [46].

### 2.4. Galectins

Galectins are members of β-galactosidase-binding proteins with pattern recognition receptors (PRRs) containing glycoconjugates with a conserved CRD for binding to the carbohydrate on bacterial membranes [60]. Three groups of galectins have so far been identified, namely, (i) a prototype containing one CRD (galectin-1, -2, -5, -7, -10, -11, -13, -14, and -15), (ii) a tandem-repeat type containing two peptide-linked CRDs (galectin-4, -6, -8, -9, and -12), and (iii) the chimera type comprising an N-terminal Pro- and Gly-rich domain fused to the C-terminal CRD (galectin-3) [87]. Their 3D structure shows two distinct CRDs, connected by a linker peptide containing 28 amino acids. The two distinct CRDs are similar to a β-sandwich structure formed by five β-sheets [9]. Nonetheless, the peptide region does not interact with the target molecules [88].

Galectin-4 (G4) sequence, isolated from the liver of striped snakehead (*Channa striatus*), was upregulated during the invasion of pathogens (i.e., *Aphanomyces invadans* and *A. hydrophila* causing epizootic ulcerative syndrome (EUS)). Furthermore, G4 exhibited weak biocidal activity against an important aquaculture pathogen (i.e., *Vibrio harveyi*). However, the biocidal activity was enhanced after tagging the C-terminal site with a pentamer oligotryptophan (W5 (G4W). Notwithstanding, G4 and G4W did not manifest any hemolytic and cytotoxicity toward peripheral blood [9]. A 1486 bp long cDNA tandem-repeat galectin-9 (RuGlec9) isolated from Korean rose bitterling (*Rhodeus uyekii*) reportedly triggered innate immunity in *R. uyekii* [59].

In addition, human cytomegalovirus and human immunodeficiency virus infections upregulated the expression of galectin-9 [89,90]. Galectin-3, a member of the chimera type family, containing a monomeric lectin with one CRD, was recently extracted from the skin and brain of turbot (*Scophthalmus maximus* L.). The authors observed down-regulation of *S. maximus* L. galectin-3 (Smgals3) in the intestine in response to *Vibrio anguillarum* and *Streptococcus iniae* infections. Further analysis revealed a strong bond with several microbial ligands suggesting that G3 may act as a cell surface docking site or a cross-linking molecule, thus facilitating adhesion [60]. In addition, the binding activity of bacteria and pathogen-associated molecular patterns (PAMPs) could be enhanced by galectin-3 (OnGal-3). OnGal-3 enhanced the binding activity by elevating the phagocytosis and respiratory burst of monocytes/macrophages, indicating that they mediated immune response via pathogen recognition and modulation of monocytes/macrophages activities [91]. To elucidate the role of galectins in infection, a recent study [24] used a zebrafish model in a challenge of infection with hematopoietic necrosis virus (IHNV), a disease-causing rhabdovirus in farmed and wild salmonid fish. Zebrafish galectin Drgal1-L2 and the chimeric-type galectin Drgal3-L1 were shown to interact directly with the glycosylated envelope of IHNV, significantly reducing viral attachment. Figure 1D shows the proposed crystal structure of the complex of Drgal1-L2 with N-acetyl-d-lactosamine at 2.0 Å resolution.

### 2.5. Galactose Binding Lectins

Galactose binding lectins (GBLs) are crucial in modulating acquired immune responses in fish [11,92,93]. In this review, we demonstrate the crystal structure of GBL using a mussel (*Mytilus californianus*) lectin (Figure 1E), which depicts α/β fold with a predominance of β structures. The lectin has specific binding to d-galactose and N-Acetyl-d-galactosamine containing carbohydrate moieties that are also inhibited by melibiose and raffinose. It can also agglutinate all types of human erythrocytes and rabbit red blood cells [25]. Some isolated GBLs express specificity for α-methyl galactose, sialoglycoproteins-like porcine and bovine submaxillary mucin and can agglutinate human rabbit, mouse, rat, and chicken erythrocytes. Examples include Ca^2+^ and pH-dependent GBL isolated from the serum of the Indian catfish (*Clarias batrachus)* [92]. Additionally, by releasing interleukin-1 like cytokines, GBLs can decrease the pathogenicity of Gram-negative bacteria (e.g., *Aeromonas* sp.) and induce the proliferation of kidney lymphocytes as observed in *C. batrachus* [92]. A GBL isolated from a marine sponge (*Chondrilla caribensis*) using affinity chromatography was identified as a homotetrameric protein formed by subunits of 15,445 Da [94]. The lectin, which showed affinity to disaccharides containing galactose and mucin, was found to agglutinate *S. aureus*, *S. epidermidis*, and *E. coli*. The lectin also expressed antibiofilm activity, owing to the α-lactose CRD.

### 2.6. Mannose-Binding Lectins

Mannose-binding lectins (MBLs) are Ca^2+^-dependent proteins that form multimeric structures with subunits. MBLs are composed of an N-terminal cysteine-rich domain, a collagen-like domain, and a C-terminal CRD. C-type lectins (CTL) with mannose-binding potentials are essential components of innate immunity in mammals owing to the stimulation of anti-inflammatory effects and augmentation of the immune response to pathogen invasion [14,79,95]. For instance, sea lamprey (*Petromyzon marinus*) plasma CTL can bind to mannose on the membrane of pathogenic *Aeromonas salmonicida* [95]. In another study, *Etroplus suratensis* lectin (Es) CTL enhanced the biocidal activity against *Vibrio parahaemolyticus* and *Aeromonas hydrophilia* and anti-cancer activity against MDA-MB-231 breast cancer cell lines [14]. Figure 1F depicts a crystal structure of aerolysin-like protein from zebrafish (*Danio rerio*) showing a β-prism lectin module followed by an aerolysin module in each subunit of Dln1. The lectin has specific binding to high-mannose glycans, which triggers drastic conformational changes in the aerolysin module in a pH-dependent manner [26].

The immunological functions of CTL4 isolated from black rockfish (*Sebastes schlegelii*) (SsCTL4) were recently reported [79]. The results showed that SsCTL4 shared a common homology with other CTLs, including CRD and disulfide bond-forming cysteine residues, as well as possesses a mannose-binding capability. CTL4 has previously been isolated from a diversity of teleost species including goldfish (*Carassius auratus*), grass carp *(Ctenopharyngod dellalla*), zebrafish (*Danio rerio*), Atlantic cod (*Gadus morhua*), lampreys (*Lampetra japonica*), rainbow trout (*Onchorhynchus mykiss*), Nile tilapia (*Oreochromis niloticus*), marine medaka (*Oryzias melastigma*), sea lamprey (*Petromyzon marinus*), snow trout (*Schizothorax richardsonii*), turbot (*Scophthalmus maximus*), and scorpionfish (*Scorpaena plumieri*) [8,16,47,95,96]. A recombinant SsCTL4 was observed to bind bacteria (e.g., *Edwardsiella tarda* and *Vibrio anguillarum*) and viral membranes in a Ca^2+^-dependent manner [79]. In addition, lily type lectins (LTLs), which share homology with d-mannose-binding lectins from plants, were reported in turbot (*Scophthalmus maximus*) (SmLTL) [47,48] and black rockfish (*S. schlegelii*) (SsLTL) [49]. LTLs are soluble PRRs with three mannose-binding sites within a three-fold internal repeat (β-prism architecture), which contains 30–99 amino acids and possesses at least one classical d-mannose- binding domain [48,49]. They exist as circa 13-kDa homodimers with a net hydrophilic structure composed of α-helices (5.36%), extended strands (39.29%), β-folds (16.07%), random coils (39.29%), and three β-folds [48]. The primary role of LTLs is to trigger the defense mechanisms in the fish mucosal immune system. The studies of Huang, Ma, Xia, Wang, Sun, Shang, Yang, and Qu [47] and Xia, Ma, Huang, Shang, Cui, Yang, and Qu [48] revealed that LTLs exert Ca^2+^-dependent hemagglutinating activity against erythrocytes and a broad range of pathogenic bacteria, including *Edwardsiella tarda*, *Vibrio anguillarum*, and *Philasterides dicentrarchi*. Rubeena et al. [97] isolated a 68-kDa mannose binding lectin from the hemolymph of Kadal shrimp (*Metapenaeus dobsoni*) (*Md-lec*). The lectin showed agglutination activity against yeast (*Saccharomyces cerevisiae*) and human erythrocytes. It also exhibited phagocytic, phenoloxidase activation, and bacterial agglutination activity against common fish pathogens, Gram-negative *Vibrio parahaemolyticus*, and *Aeromonas hydrophila*.

In another study, ficolins, Ca^2+^-independent lectins with a similar structural organization to MBL but with a fibrinogen-like lectin domain instead of CRD, were reported [98]. Ficolins can activate the complement system and the subsequent phagocytic activity against invading pathogens. They have previously been isolated from the liver of *O. niloticus* [17,99] and Atlantic halibut (*Hippoglossus hippoglossus* L.) [100]. The significant role of ficolins in the immune response has been depicted by the upregulation in the spleen and kidney of *O. niloticus* (OnMAp44) upon infection by *Streptococcus agalactiae* and *A*. *hydrophila* [99].

### 2.7. Lactose-Binding Lectins

Two structurally similar lactose-binding skin mucus lectins (24-kDa PFL-1 and 30-kDa—L-2), homologous to l-rhamnose-binding fish and sea urchin roe lectins, were found in ponyfish (*Leiognathus nuchalis)* [99].

## 3. Extraction and Purification of Lectins

### 3.1. Extraction of Lectins from Fish Muscle

A simple two-step method of extraction and purification of marine lectins was recently reported [94,101] (Figure 2). In brief, samples were finely crushed and soaked in Tris-HCl (50 mM, pH-7.6 (10 mL), NaCl (150 mM), phenylmethylsulfonyl fluoride (PMSF), Tris-buffer saline (TBS (0.1 mM), whilst the sample was stirred for 30 min at room temperature. The sample was then centrifuged at 8000× *g* for 20 min at 4 °C. The supernatant was applied to an HCL-activated Sepharose column (1.0 cm × 8.0 cm) equilibrated with TBS. The unbound protein was washed with the same buffer, and the adsorbed protein was eluted with TBS containing 200 mM α-lactose. The solvent flow rate was maintained at 1 mL/min and monitored by measuring the absorbance at 280 nm. The fractions containing purified lectins were dialyzed against distilled water, then freeze-dried and stored for further analysis.

### 3.2. Extraction of Lectins from Fish Skin Mucus

This method was developed by Sun et al. [102], using anion-exchange and gel filtration chromatography to purify the crude extract. The skin mucus of fish was collected with 200 mL of distilled water at room temperature. The collected mucus was centrifuged at 4200× *g* for 15 min at 4 °C. Mucus powder was obtained by freeze-drying the supernatant and stored at −20 °C for future use. Powdered mucus was extracted using distilled water (1/100, *w/v*) and centrifuged at 4200× *g* for 20 min at 4 °C. The precipitate was lipolyzed to get the crude extract. A desired concentration of the crude extract (10 mg/mL) was made by dissolving them in 20 mM phosphate buffer saline (pH 7.2), and 5 mL of the solution was loaded onto Φ2.6 cm × 20 cm DEAE-52 column. The lectin solution was eluted stepwise by a mixture of PBS and 0, 0.3, 0.5, and 0.7 M NaCl. The elution rate was 1 mL/min and 5 mL was collected in each test tube. Further purification of the main fractions (D-0, D-1, and D-3) was done upon dialyzed and lyophilized after spectrophotometric determination at 280 nm. D-3 was purified by gel filtration chromatography after dissolving it in 1 mL 20 mM PBS. The solution was loaded onto a Φ1.0 cm × 40 cm Sephadex G200 column. The flow rate was 0.3 mL/min, and it was collected with 2.4 mL in each test tube, and the main fraction was collected, dialyzed, and lyophilized.

### 3.3. Lectin Extraction and Purification from Crustaceans by Affinity Chromatography

This method isolated serum lectins (CqL) via affinity chromatography. Briefly, the hemolymph (serum) of the crustacean was extracted from the pericardial sinus at the first base of the abdominal segment. The hemolymph was stored at 4 °C for 24 h, followed by centrifugation for 30 min at 16,000× *g* and at 4 °C. The serum was dialyzed against phosphate buffer saline or TBS before storage at −20 °C [103,104]. The lectin was purified on 1% glutaraldehyde-fixed stroma (25 cm × 1.5 cm Sephadex G-25 column) obtained from the Wister rat or rabbit erythrocytes after hypo-osmotic treatment. Crayfish serum (1 mL) was previously applied to the column equilibrated with PBS at a flow rate of 15 mL/h. The eluted serum was washed with PBS to remove unbound materials and reduce the absorbance below 0.01 unit at 280 nm. The lectin (1.5 mL) fractions were then collected after washing with 3% acetic acid and neutralized with NaOH (1 M), dialyzed with PHB, and stored at −70 °C [103,104,105].

### 3.4. Extraction of Lectins from Bivalves via Divinyl Sulphone Activation Method

This method was developed by [106], based on the formation of adsorbents by phytohemagglutinins or lectins supporting cross-linked agarose. The divinyl sulphone activation method for lectin purification is a convenient method for the attachment of many carbohydrates. For purification of a lectin from the bivalve *M. birmanica*, [107] immobilized N-acetylglucosamine on Sepharose 4B divinyl sulphone activation was employed. The muscles of a specific organ were excised and preserved at −20 °C. About 20 g muscles were homogenized in 1 L Tris-buffered saline (TBS) (20 mM Tris–HCl, pH 7.2, 150 mM NaCl). The extract was centrifuged at 40,000× *g* for 30 min, and the supernatant was subjected to precipitation using 0–50% ammonium sulphate, followed by clarifying with further centrifugation. The precipitated protein pellet was dissolved in TBS and further dialyzed with the same buffer and filtered through a 0.2 mm filter. The solution was percolated through a 20 cm × 1 cm pre-equilibrated GlcNAc-Sepharose 4B column with TBS. The column was extensively washed with the TBS buffer until the optical density at 280 nm attained 0.005 unit, and then the bound protein was eluted with 0.4 M glucose in TBS. The purified lectins were further dialyzed by TBS and concentrated through ultrafiltration.

The affinity-purified lectin could be further purified with hydroxyapatite and ion-ex-change chromatography. Extracted fractions of lectin from the fetuin-agarose were applied to distilled water. Impurities were removed by washing with NaCl (0.5 M) until the OD at 280 nm reached 0.01 unit, and the lectins were eluted with 0.3 M potassium phosphate buffer (PPB) at pH 7.2. The eluted lectins were dialyzed against 10 mM Tris/HCl buffer (pH 8.7) and applied to a 10 mL bed volume DEAE-Bio-Gel column. The column was equilibrated and washed with the same buffer while maintaining the same pH. The bound lectin was eluted with a gradient of 0–0.4 M NaCl in the same buffer, dialyzed against PBS, and stored at −20 °C.

### 3.5. Quantification of Lectins

The carbohydrate concentration of a lectin in serum can be quantified by gas chromatography using lysine as an internal standard [108]. Lectin extracts are methanolized in methanol-HCl (0.5 M) at −80 °C for 24 h, followed by quantification of carbohydrates composition on O-methyl glycosides derivative heptafluorabutyrate. The total protein concentration in the serum lectin can be determined by the Bradford protein assay (1976) using Coomassie Brilliant Blue G-250/R-250 (absorbance measured at 595 nm) and bovine serum albumin as internal standards. Coomassie dye may be dissolved in ethanol to make it more soluble in the solution, minimizing protein aggregation during the assay [103,109]. The amino acid composition of serum lectin is determined by the liquid chromatographic system using nor-leucine as an internal standard [110]. The sample (100 mg) is hydrolyzed with HCl or water vapor in a vacuum-sealed tube for several periods (24 h, 48 h, and 72 h) at 110 °C. The samples are then analyzed by liquid chromatography consisting of two water solvents mobile phase (an aqueous buffer and acetonitrile (60%) in water) and a fixed wavelength detector (254 nm) [110]. 

## 4. Physiological Functions of Lectins in Host Body

Given their ability to selectively recognize and bind the carbohydrate moiety of glycoconjugates, one of the most prominent physiological function of lectins in marine organisms is their role as potent defense molecules where they are involved in processes such as non-self-recognition, inflammation, opsonization, phagocytosis, encapsulation, and lysis of foreign cells [111,112]. The ability of lectins to trigger multiple immunological responses is essential for autoimmunity and management of infection in fish [63,72,86,113]. For instance, the lily type lectins, CsLTL-1 and CsLTL-2, are crucial immune genes in stripped murrel *Channa striatus* for selectively recognizing and eliminating pathogens by disrupting the cell membrane [113,114]. Recently, sialic acid-binding immunoglobulin-type lectins (Siglec1, CD22, myelin-associated glycoprotein (MAG), and Siglec15) in pikeperch (*Sander lucioperca*), rainbow trout (*Oncorhynchus mykiss*) and maraena whitefish (*Coregonus maraena*) influenced the cellular reactivity against damage-associated molecular patterns (DAMPs) [63]. Brinchmann et al. [115] reported that fish mucosal lectins could trigger agglutination, inhibition of bacterial chemotaxis, endocytosis, phagocytosis, as well as inhibition of pathogens. These results suggest that fish lectins could be explored for human and veterinary medicine. The D1n1 isolated from zebrafish and other fish natterins showed a higher affinity for the envelope glycoprotein (gp120) molecule of HIV and specificity for pathogens or cancer cells, respectively [26,116,117,118]. In addition, fucose-binding lectin (DlFBL) from European bass (*Dicentrarchus labrax*) and Japanese eel (lectin 1) induced apoptosis in human cells via an adenovirus vector [119,120]. Mannose- binding lectin (MBL), galactin-1 from Atlantic cod [69,121,122,123], galectin-3, nattectin in Atlantic salmon [121,124], fucose binding lectin, and lectin-like calreticulin in European seabass [125] have previously been reported to inhibit binding or pathogens and induce phagocytosis.

Similarly, lectins participate in the innate immunity of marine hydrobionts such as clams, ascidians, sponges, and alga. A sialic acid-binding lectin from horse mussel agglutinated a variety of bacteria strains, including *Vibrio anguillarum*, *Vibrio ordalii*, *Vibrio salmonicida*, *Aeromonas* sp., *Acinetobacter* sp. and *Pseudomonas* sp. [126]. The expression of a Gal/GalNAc-specific lectin from the mussel *Mytilus trossulus* has likewise been reported to increase when the mussel is subjected to microbial challenge. Further examination shows that the lectin can induce aggregation of *V. proteolyticus* as well as bind to a number of fungi from the genera *Fusarium*, *Trichoderma*, *Haematonectria*, *Aspergillus*, and *Alternaria*, and inhibit growth and spore germination of these fungi [127]. In addition, it has been observed that a GalNAc-specific lectin in the Manila clam is upregulated in response to *Perkinsus olseni* infection. It was subsequently demonstrated that the lectin can bind hypnospores and zoospores of the parasite and that fluorescent beads coated with the lectin were actively phagocytosed by hemocytes of the clam, suggesting that the lectin may be serving as an opsonin [112]. Moreover, an increase in the expression of a tandem-repeat galectin in *Tegillarca granosa* is observed when the clam is exposed to bacterial cells or components of their cell wall [128]. In addition, a significant increase in expression of a C-type lectin (PmCTL-1) was observed in pearl oysters in response to bacterial stimulation. Purified PmCTL-1 additionally exerted strong antimicrobial activity against Gram-positive bacterial such as *M**icrococcus luteus*, *S. aureus*, and *B. subtilis* [129].

Furthermore, it has been shown that a C-type lectin from the tunicate *Styela plicata* significantly enhanced the phagocytic activity of tunicate hemocytes [130]. Furthermore, suberites lectin from the sponge *Suberites domuncula* have be shown to exhibit potent antimicrobial activity against *E. coli* and *S. aureus* [131]. Similarly, it has been reported that an N-acetylhexosamine-binding lectin in *Halichondria okadai* exhibits significant antimicrobial activity against a Gram-positive bacterium (*Listeria monocytogenes*), Gram-negative bacteria (*E. coli*, *Shigella boydii*, *Pseudomonas aeruginosa*), and a fungus (*Aspergillus niger*) [132]. Elsewhere, an endoplasmic reticulum membrane-bound lectin named Calnexin (CnX) was reported to perform a role as a pattern recognition receptor and important functions as antibacterial immunity in shrimp [133].

Besides their immune function, lectins contribute to other physiological processes including cell interaction, fertilisation, and development. In the eastern oyster, *Crassostrea virginica*, the expression of a C-type lectin (CvML) was significantly upregulated following starvation, although the exact function it plays in this situation is unknown [134]. Furthermore, fucose-binding lectin was observed to increase in the skin mucus of gilt-head bream exposed to heavy metals, suggesting that this lectin plays a role in combatting heavy metal toxicity in the fish [135]. A C-reactive protein belonging to the pentraxin superfamily has likewise been implicated in the response of *Labeo rohita* response to aquatic pollutants such as heavy metals [136]. In *Axinella polypoides*, d-galactose-specific lectins located in the spherule cells are thought to participate in the production of production of spongin [137]. In addition, some lectins can serve as antifreeze proteins. For instance, Ewart et al. reported that the type II antifreeze proteins of smelt and Atlantic herring are consistent with the CRD of C-type lectin and that it might have originated from a preexisting C-type lectin in the fish [138]. Similarly, Achenbach and Ewart identified a C-type lectin-like antifreeze protein in rainbow smelt [139].

Additionally, lectins play an important role in fertilization, embryogenesis, and morphogenesis [5]. In marine organisms such as fish and several marine invertebrates, normal development requires that the egg receives only a single sperm at fertilization. To prevent polyspermy, lectins in the cortical granule of fish eggs are released into the perivitelline space after fertilization and participating in the formation of the fertilization envelope, consequently blocking additional sperm penetration [140]. Dong et al. observed that a C-type lectin found in cortical granules of gibel carp was translocated together with cortical granules to the periphery (beneath the cytoplasm membrane) of the oocytes during maturation. The authors observed subsequently that after fertilization, cortical granules breakdown and lectins are released onto the surface and combine into the egg envelope [141]. Moreover, cortical granule lectin from unfertilized eggs of Chinook salmon is found to exhibit sperm agglutinating activity; this likely accounts for the agglutinated sperms at the base of the micropyle in the perivitelline space following fertilization [141]. In mussel and sea urchin, GlcNAc-specific lectin from *Didemnum ternatanum* is reported to promote growth and cell differentiation at the gastrula stage of the embryonic cells [142].

## 5. Other Biological Activities of Lectins

Over the years, lectins have become popular due to their therapeutic benefits, including antiviral, antibacterial, antifungal, antitumor, and anticancer activities. Some details of these activities are summarized in Figure 3 and Table 2.

### 5.1. Antibacterial Activity

The outer membrane of bacteria consists of lipoteichoic acid, lipopolysaccharides, phagocytosis receptors, and the ability to identify carbohydrate structure on bacteria. Thus, lectins inhibit the growth of bacterial cells [5,8,143]. Gram-positive bacteria have teichuronic acids, teichoic, and peptidoglycan (Figure 4A), whereas Gram-negative bacteria contain lipopolysaccharides (Figure 4B). Hydrophobic interactions, van der Waals, and Hydrogen bridges between lectin and sugar site help lectin to bind the carbohydrates, exposed on the outer membrane or cell wall of the bacteria [144]. It helps to generate a site in the cell membrane, creating its target to apoptosis by removing of intracellular components [144]. The antibacterial mechanism of lectin is shown in Figure 4C.

Lectins purified from tongue sole (*Cynoglossus semilaevis*), steelhead trout (*Oncorhynchus mykiss*), grass carp (*Ctenopharyngodon idella*), cobia (*Rachycentron canadum*), Chinese shrimp (*Fenneropenaeus chinensis*), white shrimp (*Litopenaeus vannamei*), horseshoe crab (*Tachypleus tridentatus*), bay scallop (*Argopecten irradians*), Manila clam (*Ruditapes philippinarum*), horse mussel (*Modiolus modiolus*), sponge (*Cliona varians*), marine sponge (*Chondrilla caribensis*), demosponge (*Suberites domuncula*), and sea cucumber (*Holothuria scabra*) have all been shown to exhibit anti-bacterial activity toward different bacterial species including *B. subtilis*, *E. coli* K-12, *M. luteus*, *V. anguillarum*, *Shigella* sp., *Proteus* sp., *Serratia* sp., *Streptococcus* sp., *Salmonella Minnesota* R595, *Klebsiella pneumonia*, *P. aeruginosa*, *Salmonella typhi*, *Enterobacter aerogenes*, *Aeromonas salmonicida*, and *Shewanella putrefaciens* [5,8,94,143,145]. Lectin (B-type mannose lectin) isolated from the tongue sole has been reported to reduce in vitro bacterial load of *V. harveyi* [146]. Additionally, STL lectin purified from steelhead trout has been shown to agglutinate *E. coli* K-12 and *B. subtilis* [147]. Another mannose-binding lectin (MBL) isolated from grass carp can agglutinate *S. aureus*, *V. anguillarum*, *M. luteus*, and *A. hydrophila* in a Ca^2+^-dependent manner, which reduces the survival rate of bacteria [8]. In addition, E. coli was reported to be inhibited by a tetrameric lectin isolated from cobia [10]. Furthermore, a lectin isolated from Chinese shrimp inhibited Gram-negative and Gram-positive bacteria such as *K. pneunomiae*, *E. coli*, *S. aureus*, *M. luteus*, *B. thuringiensis*, *B. megaterium*, and *B. cereus* [146]. Likewise, white shrimp C-type lectin (LvCTL3) exhibited antibacterial activity against biofilm formation by *E. coli*, *S. agalactiae*, *V. parahaemolyticus*, *S. chromogenes*, *S. hyicus*, and *S. aureus* [148]. Furthermore, tachycitin, a specific chitin-binding lectin isolated from horseshoe crab, inhibited the growth of *S. aureus*, *E. coli*, *S. typhi*, and *K. pneunomiae* [149]. Another C-type Ca^2+^-dependent lectin purified from bay scallops showed anti-bacterial activity against *M. luteus* and *V. anguillarum* bacteria [36]. Moreover, CVL, CCL, and tachylectin purified from a sponge, marine sponge, and demosponge exhibited cytotoxic effects on *E. coli* and *S. aureus* [94,131,150].

The marine sponge *Aplysina fulva* mucin-binding lectin displayed a secondary structure formed by pH-stable and thermostable β-conformations and structural resemblance to a lectin from another marine sponge *A. lactuca*. The lectin reduced the biomass of the biofilm of the bacteria *E. coli*, *S*. *aureus*, and *S. epidermidis* but was devoid of a suppressive effect on planktonic growth of Gram-negative bacteria and Gram-positive bacteria examined [64]. *N*-acetyl hexosamine-binding Japanese black sponge (*Halichondria okadai*) lectin expressed growth suppressing activity toward a fungus (*A. niger*), Gram-negative bacteria (*E. coli*, *P*. *aeruginosa*, and *S*. *boydii*), and a Gram-positive bacterium (*L. monocytogenes*). It inhibited biofilm formation in *P. aeruginosa*. The lectin entered *A. niger* conidiophores, and exhibited a pronounced inhibitory effect [132].

**Table 2 marinedrugs-20-00430-t002:** Biological activities of lectins extracted from different marine organisms.

Biological Activity	Model System	Source of Lectin	Test Types	Applied Strain	Optimum Dose	Findings	References
Antibacterial	Microorganisms and bay scallops	Bay scallop (*Argopecten irradians*) C type lectin (Ai Lec)	RT-PCR	- *Vibrio anguillarum* - *Micrococcus luteus*	50 μg/mL	Ai Lec was involved in the immune response to Gram-positive and Gram-negative microbial infection, especially *Vibrio anguillarum* and *Micrococcus luteus* in bay scallop.	[36]
	Microorganism and demosponge	Demosponge (*Suberites domuncula*) *Suberites* lectin	PCR	- *Staphylococcus aureus* - *Escherichia coli*	10 μg/mL	The lectin showed antibacterial activity against Gram-positive (*Staphylococcus aureus*) and Gram-negative bacteria (*Escherichia coli*).	[131]
	Microorganism and manila clam	Manila clam (*Ruditapes philippinarum*) Manila clam lectin (MCL-4)	Inverted microscope	- *Alteromonas haloplanktis* - *Marinococcus halophilus* - *Vibrio tubiashii*	25 μg/mL	MCL-4 had bacteriostatic properties and may contribute to the host defense mechanisms against invading microorganisms in Manila clam	[151]
	Microorganism and rabbit erythrocyte	Cobia (*Rachycentron canadum*) Tetrameric lection	Ion chromatography	- *E. coli*	250 μg/mL	The lectin showed antibacterial activity toward *E. coli*.	[10]
	Microorganism and human erythrocytes	Marine sponge (*Cliona varians*) CvL lectin	Affinity chromatography	- *Bacillus subtilis*	25 μg/mL	CvL lectin showed intense antibacterial activity against *Bacillus subtilis*.	[150]
Antiviral	Cell line, virus, and fish	Flounder (*Paralichthys olivaceus*) Galectin-1	qRt-PCR	- LCDV	50 μg/mL	Galectin-1 from flounder was able to neutralize the lymphocystis disease virus (LCDV) and exhibited anti-inflammatory activity against LCDV.	[152]
	Shrimp	Shrimp (*Penaeus monodon*) C-type lectin	PCR	- WSSV	-	A lectin domain containing PmAV protein isolated from shrimp was effective against white spot syndrome virus (WSSV)	[153]
	Virus and cell lines	Marine worm (*Chaetopterus variopedatus*) β-galactose-specific lectin (CVL)	qRt-PCR	- HIV-1	25–100 μg/mL	CVL blocked the cell–cell fusion process of the human immunodeficiency virus infected and uninfected cells with an EC_50_ of 0.07 μM and has the potential to be an anti-HIV-1 agent.	[154]
	Virus and cell lines	Sea worm (*Serpula vermicularis*) GlcNAc-specific lection (SVL)	ELISA	- HIV-1	30 μg/mL	SVL showed potential activity against human immunodeficiency virus (HIV-1) by producing viral p24 antigen, with EC_50_ values of 0.23 and 0.15 μg/mL.	[155]
Antifungal	Microbial cells	Chinese amphioxus (*Branchiostoma belcheri*) C-type lectin (AmphiCTL1)	Q-PCR	- *Saccharomyces cerevisiae*	200 μg/mL	AmphiCTL-1 lectin showed potential activity against *Saccharomyces cerevisiae*.	[153]
	Microbial cells	Orange-spotted grouper (*Epinephelus coioides*) C-type lectin (Ec-CTL)	Q-PCR	- *Saccharomyces cerevisiae*	10 μg	This lectin showed potent activity against *S*. *cerevisiae*.	[156]
	Microbial cells	Lamprey (*Lampetra japonica*) Serum lectin (NPGBP)	RT-PCR	- *Candida albicans*	10 mg/mL	The lectin showed agglutinating activities against *Candida albicans.*	[157]
Anticancer or antitumour	Virus and cell lines	Marine worm (*Chaetopterus variopedatus*) β-galactose-specific lectin (CVL)	qRt-PCR	-	25–100 μg/mL	CVL blocked the cell–cell fusion process of the human immunodeficiency virus infected and uninfected cells with an EC_50_ of 0.07 μM and has the potential to be an anti-HIV-1 agent.	[154]
	Virus and cell lines	Sea worm (*Serpula vermicularis*) GlcNAc-specific lection (SVL)	ELISA	-	30 μg/mL	SVL showed potential activity against human immunodeficiency virus (HIV-1) by producing viral p24 antigen, with EC_50_ values of 0.23 and 0.15 μg/mL.	[155]
	Tumor cell line	Chinook salmon (*Oncorhynchus tshawytscha*) Roe lectin	Microplate reader	-	-	The lectin showed intense antiproliferative activity towards human breast cancer MCF-7 cells and hepatoma Hep G2 cells.	[158]
	Cancer cell lines	Marine sponge (*Cliona varians*) CvL lectin	Flow cytometry	-	70–100 μg/mL	CvL lectin showed potential activity on K562 and Jurkat cancer cell lines.	[159]
	Cancer cells	Wheat germ (*Triticum vulgaris*) WGA lectin	Electrode array	-	100 μg/mL	A label-free electrochemical impedance spectroscopy (EIS) biosensor could be promising for the label-free profiling of the glycan expression of cancer-related glycoproteins in the early stage of a cancer diagnosis.	[160]

The transcription of shrimp (*Litopenaeus vannamei*) perlucin-like protein, a C-type lectin, was upregulated following challenge with the Gram-negative bacteria *V*. *anguillarum* and *V. parahaemolyticus*. The recombinant lectin showed binding activity to peptidoglycan and lipopolysaccharide, albeit with dissimilar affinities, and binding to Gram-negative bacteria (*V. anguillarum* and *V. parahaemolyticus*) as well as Gram-positive bacteria (*B*. *subtilis* and *S*. *aureus*). It agglutinated *V. anguillarum* and *V. parahaemolyticus*, but not *B. subtilis* and *S. aureus*. Expression of the genes of the antimicrobial peptides ALF1, ALF2, ALF3, and crustin, as well as those of the phagocytosis-related genes dynamin, mas-like protein and peroxinectin were downregulated after lectin gene knockdown by RNA interference [161,162]. 

Kuruma shrimp (*Marsupenaeus japonica*) galectin recognized Gram-negative as well as Gram-positive bacteria. It was capable of crosslinking pathogenic microorganisms to hemocytes, promoting phagocytic activity of the hemocytes and eliminating the pathogen from the circulation. Eastern oyster (*Crassostrea virginica*) galectins recognized a considerable number of different bacteria and the parasitic protozoan *Perkinsus marinus*, and displayed adhesive and phagocytic activities toward the parasite. Softshell clam (*Mya arenaria*) galectin recognized *P. chesapeaki* and *P. marinus*. Nevertheless, the Perkinsus parasites have means to evade immune functions of the galectins [163]. An augmented expression of freshwater prawn (*Macrobrachium rosenbergii)* l-type lectin was detected in hemocytes following exposure to White spot syndrome virus, *S*. *aureus* and *V*. *parahaemolyticus.* The recombinant form of the lectin demonstrated binding and agglutinating activities toward all bacterial species examined, and also binding activity to bacterial surface peptidoglycans and lipopolysaccharide. The recombinant lectin exhibited in vitro growth inhibitory activity toward microbes and promoted in vivo clearance of pathogenic bacteria and suppressed replication of white spot syndrome virus [50].

Mud crab (*Scylla paramamosain*) C-type lectin bound pathogen-associated molecular patterns (glucan, lipoteichoic acid, lipopolysaccharide, and peptidoglycan) and agglutinated a variety of bacteria. The recombinant lectin promoted hemocyte encapsulation activity, suppressed the level of lipopolysaccharide and increased *S. paramamosain* survival following challenge with *V. alginolyticus*, although it did not kill the bacterium [44]. The recombinant form of swimming crab (*Portunus trituberculatus*) mannose-binding lectin showed Ca^2+^-dependent binding and agglutinating activities toward both yeast and bacteria and inhibitory activity toward the Gram-positive and Gram-negative bacteria examined. The lectin level in hemocytes was upregulated following exposure to *M*. *luteus*, *Pichia pastoris* and *V*. *alginolyticus* [164]. The mud crab (*Scylla paramamosain*) hepatopancreas C-type lectin showed Ca^2+^-dependent agglutinating activity toward three each of Gram-negative bacteria and Gram-positive bacteria [164]. The mRNA expression level of mud crab *(Scylla paramamosain)* galectin rose abruptly upon challenge by *V*. *alginolyticus.* The lectin bound bacterial cell wall lipopolysaccharide, and agglutinated three Gram-positive (*B.*
*aquimaris*, *M*. *lysodeik* and *S*. *aureus*) and three Gram-negative (*Aeromonas hydrophila*, *Chryseobacterium indologenes* and *V. alginolyticus*) bacteria [165].

Recombinant swimming crab (*P. trituberculatus*) l-type lectin agglutinated all of the three Gram-positive and Gram-negative bacteria examined. The lectin exhibited agglutinating activity toward red blood cells and lipopolysaccharide-binding activity. The expression level of the lectin was increased in hemocytes exposed to *Vibrio alginolyticus*, signifying its protective role during bacterial infection [166]. Gazami crab (*Portunus trituberculatus*) C-type lectin is characterized by the possession of only one carbohydrate-recognition domain with a conserved QPD motif. The mRNA transcript of the lectin which was found mainly in the gut rose dramatically after exposure to pathogenic microbe. It exhibited agglutinating activity toward Gram-positive bacteria (*M*. *luteus* and *S*. *aureus*) and Gram-negative bacteria (*P*. *aeruginosa* and *V*. *alginolyticus*) but bound yeast only weakly. The recombinant lectin exhibited antibacterial activity and increased elimination of *V. alginolyticus*. The transcription of three antimicrobial peptides (PtALF4-7, PtCrustin1, and PtCrustin3), three prophenoloxidase system-related genes (PtproPO, PtcSP1, and PtPPAF), two phagocytosis genes (PtMyosin and PtArp), and PtRelish was downregulated by knockdown of the lectin gene. PtJNK, PtPelle, and PtTLR were expressed at higher levels [167].

Sea urchin (*Pseudocentrotus depressus*) mannose-binding lectin in various sea urchin tissues, combats body surface microbes on the sea urchin. The lectin demonstrated agglutinating activity toward Gram-positive *Lactococcus garvieae* [42]. The gene encoding of the razor clam (*Sinonovacula constricta*) C-type lectin has an N-terminal signal peptide, 4 C-type carbohydrate recognition domains and a C-terminal transmembrane region. The binding and agglutinating activities of the lectin and the hemocyte phagocytosis enhancing activity toward bacteria, and the Ca^2+^-dependence of carbohydrate binding activity of the lectin, were demonstrated. The hepatopancreas showed the highest level of expression of the lectin although it was detected in nearly every tissue. Hemocyte expression of the lectin escalated subsequent to challenge with bacteria [168].

The recombinant forms of two Manila clam (*Venerupis philippinarum*) C-type lectins (rVpClec-3 and rVpClec-4) agglutinated *V*. *anguillarum*, *V. harveyi*, and *V. splendidus* and upregulated phagocytic activity of hemocytes. However, rVpClec-3 was devoid of agglutinating activity toward *Aeromonas hydrophila* or *Enterobacter cloacae* [41]. Snail (*Hemifusus pugilinus*) C-type lectin agglutinated a diversity of Gram-positive and Gram-negative bacteria [40]. The recombinant turbot (*Scophthalmus maximus*) C-type lectin combined with peptidoglycan, lipoteichoic acid, and *lipopolysaccharide*, agglutinated five different bacteria in the presence of calcium [169]. The 237-amino acid oyster (*Crassostrea gigas)* C-type lectin exhibited agglutinating and growth-suppressing activities toward *V*. *alginolyticus*. Its expression, which was observed predominantly in the digestive gland, was elevated after exposure to *V. alginolyticus.* The subcellular locations of the lectin suggest that it has multiple roles in the immune response. The lectin may be implicated in recognition of food particles [170].

From the oyster *Crassostrea gigas*, *Cg*CLec-TM1 (a novel C-type lectin with a transmembrane domain), *Cg*ERK (the ERK homolog) and *Cg*GSK3β (GSK3β homolog) were identified (reference). The C-type lectin showed binding activity, through its carbohydrate recognition domain, toward *E*. *coli* and *V*. *splendidus*. *Cg*ERK was activated by the C-type lectin through phosphorylation. The activated *Cg*ERK interacted with *Cg*GSK3β which was phosphorylated at the Ser9 position, and stimulated *Cg*IL-17-1 and *Cg*IL-17-5 expression. The *Cg*ERK-*Cg*GSK3β interaction, and *Cg*GSK3β phosphorylation, was inhibitable by PD98059 (ERK inhibitor) to decrease *Cg*IL-17-1 and *Cg*IL-17-5 expression. Oyster *Cg*GSK3β was a novel *Cg*ERK substrate. In the oyster, the CLec-TM1-ERK-GSK3β signaling pathway was stimulated by *V. splendidus* and, subsequently, *Cg*IL-17 formation was stimulated [162]. CgCLec, an oyster *Crassostrea gigas* C-type lectin with a CCP (complement control protein) domain, capable of recognizing bacteria and pathogen-associated molecular patterns, was implicated in activating the complement system by binding oyster mannose-binding lectin-associated serine protease-like protein (CgMASPL-1) in order to facilitate CgC3 cleavage. The levels of antibacterial peptides, cytokines, and hemocyte phagocytic ratios in oysters with knockdown of CgCLec-CCP-, CgMASPL-1-, or CgC3-, declined following exposure to the bacterium *V. splendidus*. After activation, CgC3 introduced holes in the bacterial envelopes and reduced bacterial survival [162].

Intelectin is a soluble galactofuranose-binding lectin that exists in species ranging from amphioxus to human. Domain 5 (aa 203–302) of Chinese amphioxus (*Branchiostoma belcheri* tsingtauense) interlectin-bound peptidoglycan or lipopolysaccharides and agglutinated bacteria as efficiently as the full-length protein. Four amino acids mediating calcium-binding (G54-G55-G56-E91) were identified by hemagglutination assay. A striking functional conservation of Domain 5 was detected in zebrafish intelectin 1 [171].

The recombinant form of the Wuchang bream *Megalobrama amblycephala* intelectin expressed binding and agglutinating activities toward bacteria and lipopolysaccharide-binding activity and the interlectin was detected at an elevated level in macrophage-like kupffer cells in the liver, which demonstrated increased microbial killing after a challenge with *Aeromonas hydrophila*. The recombinant intelectin suppressed proliferation of cancer cells [172].

A rise in the expression of spotted knifejaw (*Oplegnathus punctatus*) roe C-type lectin occurred after exposure of the fish to Gram-negative *Vibrio anguillarum.* The recombinant form of the lectin demonstrated binding and agglutinating activities toward two Gram-positive and four Gram-negative bacteria, and in addition, binding activity to bacterial surface peptidoglycan and lipopolysaccharide [173]. A tandem-repeat galectin-1 from *Apostichopus japonicus* with broad PAMP recognition pattern bound and agglutinated Gram-positive bacteria (*M*. *leteus*), Gram-negative bacteria (*E. coli* and *V. splendidus*), and fungi (*P*. *pastoris*), and demonstrated suppressive activity on *E. coli* and *V. splendidus* [173].

The 35.8-kDa rockfish galectin-8, with both N and C-terminal carbohydrate binding domains, agglutinated both Gram-positive as well as Gram-negative bacteria, including *E. coli*, *Lactococcus garvieae*, *S*. *iniae*, *S*. *parauberis*, *V*. *harveyi*, *V*. *parahaemolyticus*, and *V*. *tapetis*. Exposure to *S*. *iniae* and lipopolysaccharide elevated mRNA transcription levels of the galectin in the head, kidney, liver, and spleen, indicating the immunoregulatory and anti-infective role of the galectin in the rockfish [174].

Recombinant Nile tilapia (*O. niloticus*) galectin-4 homolog exhibited binding and agglutinating activities toward *A. hydrophila* and *S. agalactiae*. The galectin stimulated expression of cytokines and enhanced antibacterial activity of monocytes/macrophages. Overexpression of the galectin served to prevent *O. niloticus* from infection by *S. agalactiae* by regulating the inflammatory response. The galectin plays a role in pathogen recognition and opsonization to combat bacterial invasion [91]. The 145-amino acid Nile tilapia (*O. niloticus*) galectin-related protein B exhibited a conserved carbohydrate recognition domain with partial sugar binding sites (N-R, V-N and W-E) and an amino acid sequence homology to galectin-related protein B from other fish species. The lectin in healthy tilapia showed a wide tissue distribution and presence in monocyte/macrophages. The levels of transcripts of the lectin were heightened in the head, kidney, liver, spleen, and monocyte/macrophages confronted with *S*. *agalactiae.* The recombinant lectin demonstrated binding and agglutinating activities toward bacteria. The lectin modulated inflammatory factors and possessed antibacterial activity [175]. The results indicate that the lectin protects the Nile tilapia against bacterial infection.

Galectin-3C, the C-terminal half of galectin-3 isolated from Atlantic salmon skin mucus, exhibited agglutinating activity toward Gram-negative *Moritella viscosa*. Elevated expression of galectin-3 mRNA was noted in the gills and skin, followed by muscle, hindgut, spleen, stomach, foregut, head kidney, and the liver. Galectin-3C treatment altered the amount of three ribosomal proteins L7/12, S2, and S13 and multidrug transporter in *M. viscosa* [74]. The mannose-binding skin mucus pufflectin of the pufferfish *Takifugu rubripes*, which manifested sequence resemblance to monocot mannose-binding lectins but not to animal lectins, showed binding activity to the trematode parasite *Heterobothrium okamotoi*. The ponyfish *(Leiognathus nuchalis*) lectin is homologous to some rhamnose-binding fish roe lectins [176]. The levels of expression of two obscure puffer (*Takifugu obscurus)* C-type lectins were upregulated in response to infection with *A. hydrophila* and *V. harveyi*. The recombinant forms of the lectins showed Ca^2+^-dependent agglutinating activity toward Gram-positive and Gram-negative bacteria, exhibited binding to bacterial peptidoglycan and lipopolysaccharide, and manifested in vitro growth inhibitory activity on four types of bacteria [177]. Calreticulin with lectin-like and immune activities is a highly conserved protein. The obscure puffer (*Takifugu obscurus*) calreticulin homolog comprises an N-domain, a P-domain, and a C-terminal domain which are structurally conserved. Obscure puffer calreticulin homolog forms a separate cluster with three other pufferfish (*T. bimaculatus*, *T. flavidus*, and *T. rubripes*) calreticulins in the phylogenetic tree. Following a challenge of the obscure puffer with *A. hydrophila*, *Edwardsiella tarda*, and *V. harveyi.*, the renal and splenic levels of calreticulin homolog mRNA were elevated. The recombinant calreticulin domain of the calreticulin homolog showed binding activity toward *A. hydrophila*, *E. tarda*, and *V. harveyi* and bacterial cell wall peptidoglycan and lipopolysaccharide. Obscure puffer calreticulin homolog manifested agglutinating and antimicrobial activities toward a variety of microbes [178]. The mandarin fish (*Siniperca chuatsi*) galectin-8 exhibited growth inhibitory activity on *S*. *agalactiae* and *Flavobacterium columnare*, and galectin-9 also inhibited the growth of *Edwardsiella piscicida* [179].

An elevated expression of half-smooth tongue sole (*Cynoglossus semilaevis*) collection 11, a novel complement-associated pattern recognition molecule, was detected in the blood, gills, head, kidney, and spleen, 6 to 24 h following challenge with *V*. *anguillarum.* The blood level of the C-type lectin CsCL-11 was upregulated. Moreover, by binding to various bacteria, recombinant CsCL-11 (rCsCL-11) expressed in human embryonic kidney cells HEK-293 T cells expressed strong bacterium binding and antibacterial activities [180]. Recombinant tongue sole *Cynoglossus semilaevis* galectin-8 bound to various Gram-positive and Gram-negative bacteria and the binding was inhibitable by fructose, galactose, lactose, and trehalose. The galectin killed some Gram-negative bacteria by disrupting the membrane [181]. The recombinant form of tandem-repeat galectin-9 from the Japanese flounder (*Paralichthys olivaceus*) bound to peptidoglycan and lipopolysaccharide, and agglutinated Gram-positive as well as Gram-negative bacteria [182].

F type lectin from the fish *Trematomus bernacchii* demonstrated bacterium agglutinating activity at a low temperature [71]. Following exposure to pathogenic microbes (*E*. *piscicida*, *S*. *iniae*, or red sea bream iridovirus), the level of mRNA expression of red seabream (*Pagrus major)* galectin *Gal-9* was heightened in most immune tissues, indicating the role of the galectin in the red sea bream immune system [183]. The recombinant form of galectin-8 from the Japanese flounder *Paralichthys olivaceus* bound to Gram-negative bacterial cell wall lipopolysaccharide and Gram-positive bacterial cell wall peptidoglycan and agglutinated the bacteria studied [184]. Recombinant galactose-binding lectin from the Japanese sea bass (*Lateolabrax japonicas*) bound to monosaccharides and polysaccharides, and both native and recombinant forms of the lectin Ca^2+^ dependently agglutinated three Gram-positive bacteria (*M*. *luteus*, *S*. *aureus*, and *S*. *iniae*) and four Gram-negative bacteria (*A. hydrophila*, *E. tarda*, *V. anguillarum*, and *V. harveyi*) in vitro. After infection with *V. harveyi*, *the* recombinant lectin reduced bacterial titers in the blood, kidney, liver, and spleen and enhanced fish survival [185].

### 5.2. Anti-Viral Activity

Some lectins exhibit a specific binding behavior to carbohydrate structures which is the key component of the gp120 on the viral envelope that renders to inhibit the conformational change that keeps it in an inactive state. It is speculated that these lectins could be used to develop antiviral drugs [143,186]. Galectin-1, a Ca^2+^ ion independent lectin, bound to β-galactosidase and neutralized lymphocystis disease virus (LCDV). It also inhibited the cytopathy of the infected cells against inflammation [152]. A shrimp protein containing a C-type lectin-like domain (6.25 µg/mL) encoded by PmAV effectively suppressed viral replication compared to the control group [153]. In vitro experiments showed that *Serpula vermicularis* lectin (SVL) inhibited syncytium formation in C8166 cells infected by human immunodeficiency virus type 1 (HIV-1) in a dose-dependent manner. Similarly, SVL (0.23 μg/mL) decreased the production of HIV-1 p24. A positive control experiment with azidothymidine (AZT) showed similar results [155]. High-mannose specific lectins, namely His-rKAA-1, KAA-1, and KAA-2, decreased the infectivity of HIV-1 at low nanomolar IC_50_ of 12.9 ± 2.2, 9.2 ± 2.2, and 7.3 ± 1.9 nM, respectively, by binding to gp120, thus preventing the virus from entering into host cells [187]. Similarly, high mannose-specific lectin (KAA-2) isolated from the red alga *Kappaphycus alvarezii* showed significant virucidal activities against all influenza viruses (A/WSN/33 (H1N1), A/FM/1/47 (H1N1), A/Kyoto/1/81 (H1N1), A/Bangkok/10/83 (H1N1), A/Beijing/262/95 (H1N1), A/Aichi/2/68(H3N2), A/Udorn/72 (H3N2), A/Philippines/2/82 (H3N2), B/Ibaraki/2/85) assessed except PR8/34 (H1N1), which showed no sensitivity even at a higher dose (70 nM). The authors further demonstrated that 200 nM KAA-2 completely prevented viral entry into Madin–Darby canine kidney (MDCK) cells, as shown by the non-detection of antigens similar to the negative control. However, in a positive control experiment with Amantadine [131], A/Oita/OU1 P3-3/09 (H1N1) infected and proliferated in the host cells as evidenced by the detection of viral antigens [188]. Microvirin (MVN) extracted from *Microcystis aeruginosa* showed 33% similarity to an anti-HIV protein cyanovirin-N (CV-N), which exerted antiviral activity by inhibiting syncytium formation in HIV-1-infected T cells and healthy CD4^+^ Tcells, thus preventing the binding and transmission of HIV-1 to CD4^+^ T cells. Marine derived lectin, Griffithsin (GRFT), exerted a strong antiviral activity against several viruses, including HIV, hepatitis C virus, severe acute respiratory syndrome coronavirus (SARS-CoV), and Ebola virus in the serum of rodents [157,189,190,191,192,193]. Furthermore, GRFT inhibited the binding of gp120 to receptor-expressing cells as well as 2G12 and 48d monoclonal antibody in a glycosylation-dependent manner. Interestingly, GRFT binding to gp120 was unaffected by galactose, xylose, fucose, N-acetylgalactosamine, or sialic acid-containing glycoproteins. On the other hand, monosaccharides such as glucose, mannose, and N-acetylgluosamine hindered the binding of GRFT to soluble gp120 [157]. On the other hand, monosaccharides (i.e., glucose, mannose, and Glc-NAc) decreased the virucidal activity [157]. It was further reported that GRFT at the concentrations of 181 µg/mL (13.9 nM) and of 644 µg/mL (49.5 nM) showed 50% and 90% inhibition, respectively, against JFH1 HCV in a cell culture model, whereas mutated GRFT at carbohydrate bindings sites (CBS) showed no antiviral activity [192].

#### 5.2.1. Coronavirus

There are four genera of coronaviruses (CoVs): alpha-, beta-, gamma-, and delta-coronavirus. Alpha-coronavirus and beta-coronavirus are transmitted to mammals and cause SARS-CoV and Middle East Respiratory Syndrome (MERS-CoV) [194]. Gamma-coronavirus and delta-coronavirus infect both mammals and birds. Following the outbreak in Wuhan, China, in late 2019, a novel coronavirus different from SARS-CoV and MERS-CoV was discovered. In view of its close genetic relationship to SARS-CoV, it was designated as SARS-CoV-2 [195,196]. SARS-CoV-2 virion has a diameter of 50–200 nm, and possesses a 26–32 kilobase RNA genome with a positive sense. The genome codes for four predominant structural proteins, including envelope (E), membrane (M), nucleocapside (N), spike (S), and a few non-structural proteins. The spike protein plays a central role in binding to the host cell receptor, membrane fusion, viral internalization, and host tropism. The envelope protein is implicated in the morphogenesis, release, and pathogenesis of the virus, whereas the membrane protein binds to the nucleocapside and facilitates viral assembly and budding [195,196,197,198,199,200,201]. Coronavirus infection is initiated by the interaction of the viral protein with the host cellular membrane. The angiotensin-converting enzyme 2 (ACE2) in the host cell is the targeted membrane protein, and SARS-CoV-2 has a higher affinity for ACE2 than any other coronavirus [195,196,199].

The mechanism of viral entry, replication, transcription, and assembly of the virus particle in the human cell is shown in Figure 5. The S proteins attach to the ACE2 receptors, which are found on the lung cells. The attachment results in endocytosis, which is the primary ingestion mechanism of the virus by the human cells. Once inside the cytoplasm, SARS-CoV-2 virion causes membrane fusion through receptor binding and induces conformational changes in S protein followed by proteolysis with the aid of intracellular proteases and activates membrane fusion mechanism within endosomes. The endosome releases the virus to the cytoplasm, and uncoating of viral nucleocapside is initiated via proteasomes which hydrolyze endogenous proteins along with exogenous ones. Finally, the viral genetic material, a single-stranded RNA, is released into the cytoplasm. Replication and transcription are processed by the replication-transcription complex (RTC), where the viral genome is encoded.

#### 5.2.2. Lectins as Potential Inhibitors of SARS-CoV-2

Lectins may suppress SARS-CoV-2 replication through interaction with envelope glycoproteins of the virus ensuing in viral clumping, thus inhibiting infection of the host (Figure 6). The antiviral activity of lectins is suggested to the targeting of multiple stages of the viral life cycle, such as their entry, attachment, and replication mechanisms [202]. For instance, lectins may induce conformational rearrangement of the viruses by binding with the viral envelope glycoprotein, thus inhibiting the fusion with the host receptor. Fish lectins exert an inhibitory action on SARS-CoV-2 hemagglutinin-esterase [203].

C-type lectins such as CD209/DC-SIGN and CD209L/L-SIGN proteins possess different receptors for cell adhesion and pathogen recognition which promote cellular interactions and recognize a diversity of pathogenic microorganisms, including SARS-CoV-2 [204]. Lectins inhibit spike proteins and hence viral entry to the host [205]. *O. niloticus* lectins augmented interferon-γ (IFN-γ) formation, a potential immunomodulator of Th-1 type immune response which may combat the coronavirus. Furthermore, cytokine production (controlled) may be heightened, reinforcing the host immune response to combat SARS-CoV-2 [16]. GRFT inhibited other CoVs which infect man, other mammalian, and avian species [206]. Approaches have been proposed to upgrade the anti-CoV effectiveness of carbohydrate-binding agents; and molecular docking has been employed to investigate molecular interactions between carbohydrate-binding agents (lectins and Pradimicin-A) with Man9 for anti-CoV and anti-SARS-CoV- 2 repurposing [207]. The replication of enveloped viruses is suppressed in the presence of sulphated algal polysaccharides. The algal lectin griffithsin, the phycocolloid carrageenan, and the sulfated algal polysaccharides fucoidans and ulvans could serve as anti-SARS-CoV-2 therapeutics [208].

### 5.3. Anti-Fungal Activity

Although there are several lectins identified from fish, only a few manifested anti-fungal effects. Lectins usually inhibit fungal growth by changing the structure of the fungal cell membrane or binding with carbohydrates of the fungal cell wall, thus interaction with the glucans and peptidoglycans leads to killing of the fungi by cell wall permeabilization [5,143,145,209]. Among the various fish lectins, C-type lectins have strong activity against fungi [143]. AiCTL-7, a C-type lectin isolated from bay scallop (*Argopecten irradians*), was shown to inhibit the growth of the methylotrophic yeast *P*. *pastoris* [210]. Another C-type lectin (Ec-CTL) isolated from an orange-spotted grouper (*Epinephelus coioides*) binds to *S. cerevisiae* in a Ca^2+^-dependent manner to cause inhibition of growth [211]. Other lectins obtained from mussels (*Crenomytilus grayanus* and *Mytilus trossulus*) have the potential to inhibit conidia germination, especially in *Trichoderma*, *Fusarium*, *Haematonectria*, *Alternaria*, and *Aspergillus* strains due to their anti-fungal effect [127,212]. Even though fish lectins have potential anti-fungal activity, few researchers have exploited them to develop drugs for humans. Thus, further studies on the structure–function relationship, clinical trials, and molecular mechanisms of action might help researchers to unveil the toxicity of lectins and therapeutic effects that are essential to developing proper doses of lectins for consumption as human medicine [212,213,214].

The recombinant form of the variant of the red algal lectin Griffithsin recalcitrant to oxidation, designated as Q-Griffithsin, showed binding activity to *Candida albicans* cell wall α-mannan, destroyed cell wall intactness and triggered production of reactive oxidative species, leading to cell death. Q-GRFT exerted a growth inhibitory action on *C. glabrata*, *C. krusei* and *C. parapsilosis*, and mildly inhibited some multidrug-resistant and pandrug-resistant *C. auris* strains. Q-GRFT regulated expression of cell stress response genes involved in cell cycle progression and reactive oxidative species production [215]. Snail (*Hemifusus pugilinus*) C-type lectin inhibited growth of the pathogenic fungi *Aspergillus flavus* and *A*. *niger* at concentrations of 25 and 50 μg/mL [40].

### 5.4. Activity Related to Homeostatic Maintenance of Intestinal Microbiota

Kuruma shrimp (*M. japonicus*) intestinal C-type lectin enhanced intestinal bacterial establishment in the gut and homeostatically regulated the intestinal microbiota. Silencing of the lectin gene brought about tissue injury, intestinal dysbacteriosis, and mortality. The lectin facilitated intestinal bacterial biofilm production due to a bacteria recognition lectin domain and a coiled coil region for dimerizing CTL33 and bacterial cross-linking into a complex resembling a biofilm [216].

### 5.5. Anti-Cancer or Anti-Tumor Activities

Lectins are widely involved in mitogenic cytotoxicity, cell adhesion, apoptosis, and premalignant, and malignant cell recognition [143,217,218]. In addition, malignant cells frequently change glycosylation patterns unlike normal cells. Nonetheless, and despite the multiple mechanisms of action, lectins have an essential role in hindering the growth and proliferation of malignant and premalignant cells both in vivo and in vitro [5,143]. Lectins isolated from chinook salmon (*Oncorhynchus tshawytscha*), bighead carp (*Aristichthys nobilis*), Atlantic cod (*Gadus morhua*), tongue sole (*Cynoglossus cynoglossus*), sea bass (*Dicentrarchus labrax*), catfish (*Silurus asotus*), Japanese eel (*Anguilla japonica*), sea mollusk (*Crenomytilus grayanus*), sponge (*Cliona varians*), demosponge (*Halichondria okadai*), and mussel (*Mytilus galloprovincialis*) were found to be effective against colon tumor, thyroid tumor, breast tumor, lung cancer, and liver cancer [5,143,218]. Furthermore, a lectin obtained from chinook salmon roes showed potent anti-proliferative activity against human breast cancer cells (MCF-7 and Hep G2 cells) and facilitated nitric oxide production from mouse peritoneal macrophages, which exhibited antiproliferative activity toward tumor cells [158].

Likewise, a GANL lectin purified from bighead carp exhibited anti-tumor activity, independent of Ca^2+^, in HeLa tumor cell lines [219]. Galectin-3 and galectin-1, found in Atlantic cod, tongue sole and sea bass, was observed to usually bind to glycoconjugates of the malignant cells of the blood vessels, a mechanism that portrays its potential for the treatment and diagnosis of cancer [122,220,221]. Furthermore, Gb3, a lectin isolated from catfish, can shrink Burkitt’s lymphoma cells by stimulating the potassium channel Kv1.3 [222]. Lectin-1 purified from Japanese eel exhibited cytotoxicity activity in different human lung and liver cancer cell lines [119,120]. Lectin CGL collected from sea mollusk has anti-cancer activity against breast cancer cells by binding with globotriose [223]. Human leukemia cells are susceptible to CvL lectin isolated from a sponge (*Cliona varians*) and lectin of sponge induced to release cathepsin B that leads to induction of apoptosis. HOL-18 purified from demosponge was reported to show cytotoxic activity in Jurkat leukaemia and erythroleukemia cells (T cells and K562) [159,224]. Additionally, MytiLec, a 17-kDa lectin isolated from a mussel, can bind with globotriose and has the potential to suppress the growth of human lymphoma cells by compromising the integrity of the cell membrane [225].

Red alga (*Kappaphycus striatus)* lectin displayed antiproliferative activity against AGS gastric cancer cells, Hela cervical cancer cells, HT29 colorectal cancer cells, MCF-7 breast cancer cells, and SK-LU-1 lung denocarcinoma cells, with IC50 values in the range of 0.8–1.94 µM. Yeast mannan, which inhibited the lectin, abolished the anticancer effect of the lectin [226]. The IC50 values of the cytotoxic activity of acetyl hexosamine-binding Japanese black sponge (*Halichondria okadai*) lectin on breast cancer MCF-7 cells and T47D cells and cervical cancer HeLa cells, were 52, 63 and 40 μg/m, respectively. MAPK phosphorylation and apoptotic caspase-3 activation were involved in the mechanism of action of the lectin on HeLa cells [132]. Genes encoding sponge (*Aphrocallistes vastus*) C-type lectin enhanced the cytotoxic action of oncolytic vaccinia virus (oncoVV) toward tumor cells. The complex formed between the lectin and the virus exerted an in vitro as well as an in vivo growth inhibitory action on Hela S3 cervical cancer cells [227].

There are different families of mytilid lectins such as R-type lectins and mytilectins sharing a common β-trefoil fold structure but with distinct specificities of glycan binding. The 15-kDa purplish bifurcate mussel (*Mytilisepta virgata*) lectin SeviL was classified as an R-type (Ricin B-type) lectin, due to the existence of the typical QxW motif. SeviL induced apoptosis in BT474 human breast cancer cells, Caco2 colonic cancer cells, and HeLa cervical cancer cells with cell surface asialo-GM1 oligosaccharides. This cytotoxicity was nullified by antibodies against asialo-GM1 oligosaccharide. Activation of caspase-3/9 and kinase MKK3/6, p38 MAPK were involved in the mechanism of action of the lectin on HeLa cells [28]. 

### 5.6. Lectins as an Immunity Enhancer

Lectins are able to trigger an immune response via several mechanisms. First and foremost, they act as opsonins, which enhance phagocytosis, activation of the complement pathway, modulation of inflammation, and inhibition of a plethora of microbes [115,228,229,230]. Lectins can also activate intracellular signaling pathways to increase NF-κB and type I interferon production, augment inflammasome activation, and induce a wide range of cellular and immunological responses [231]. Furthermore, lectins may bind to microbes and enhance agglutination, complement-mediated neutralization, and killing of infected cells [5,115,230]. They also regulate the recognition, adhesion and migration of cells, morphogenesis, glycoprotein synthesis, and other blood proteins [232]. Moreover, the major histocompatibility complex (MHC) molecules, which play a pivotal role in the immune system by presenting peptide molecules for recognition by T-cells, are influenced by lectins [233]. It was also reported that, in microbes, lectins spliced essential RNA, enhance the folding of important proteins, transform molecules, and mediate cellular assembly [115]. Moreover, through myeloid cell recognition, it can mediate the uptake of microbes through phagocytosis and induce adaptive immune responses (Figure 7 and Figure 8) [228]. According to [234], lectins can influence the following phenomena: amplification of human leucocyte antigen (HLA) class II expression; stimulation of T-cell proliferation; stimulation of IFN-g; mediation of abnormal expression of ICAM in T-cells; and stimulation of inflammatory cytokines production (i.e., IL-1, TNFα).

F-type lectins possess a fucose recognition domain with a “F-type” jellyroll fold and unique sequence motifs for binding calcium and carbohydrate. These lectins play a role in diverse processes including fertilization, innate immunity, adhesion of microbes, and pathogenesis [235]. F-type lectin (fucolectin) isoforms (AjFTL-1 and AjFTL-2) from the sea cucumber (*Apostichopus japonicus*) exhibit disparate immune roles during bacterial infection. The *Vibrio splendidus* infection in vivo more strongly upregulated the mRNA transcripts of AjFTL-1 compared to AjFTL-2. However, only AjFTL-2 was upregulated by lipopolysaccharide stimulation in vitro. Silencing AjFTL-2 using siRNA downregulated the AjNOS transcript, and AjNOS expression was upregulated by recombinant AjFTL-2. AjNOS expression was unaffected by the loss- and gain-of-function of AjFTL-1 [236].

Recombinant sea cucumber *A. japonicus* C-type lectin rAjCTL-2 displayed binding activity towards a number of microbes including *Bifidobacterium breve*, *S*. *aureus*, *V*. *anguillarum*, *V*. *splendidus*, and *Yarrowia lipolytica* with the highest potency to *B. breve* [236]. ß-Galactoside-binding galectins, which possess a unique sequence motif in the carbohydrate recognition domain, display a myriad of roles encompassing recognition of endogenous carbohydrate ligands in embryo initiation and development and early development, adipogenesis, tissue repair, immunoregulation, and cancer. They have adhesive, phagocytosis enhancing, and cidal effects toward pathogenic microbes, and facilitate encapsulation, autophagy, and pathogen clearance. Nevertheless, some pathogenic microorganisms may evade galectin-mediated innate immune mechanisms [237].

### 5.7. Other Potential Applications

Lectin from the marine alga *Solieria filiformis* exerted an antidepressant-like effect, but was devoid of anxiolytic-like and psychostimulant effects. The antidepressant-like effect was inhibited by a dopamine D1 as well as a dopamine D2 receptor antagonist indicating involvement of the dopaminergic system [238]. Enhanced expression of the lectins (CvML3912 and CvML3914) in the oyster (*Crassostrea virginica)* feeding organs was detected after starvation. The two cognate transcripts were downregulated by gene silencing using Short Dicer-substrate small interfering RNA (DsiRNA) targeting these two lectins and food sorting ability. Thus, the food particle sorting role of these metazoan mannose/glucose-binding proteins was demonstrated [239]. Juvenile coral, *Acropora tenuis*, secrete an *N*-acetyl-d-glucosamine-binding lectin to attract dinoflagellate (*Symbiodinium tridacnidorum*). The ranking of chemotactic activity is as follows: CS-161 (*Symbiodinium tridacnidorum*) and CCMP2556 (*Durusdinium trenchii*) >CCMP1633 (*Breviolum* sp.), but there was negligible or no attraction towards CCMP421 (*Effrenium voratum*), CS-156 (*Fugacium* sp.), FKM0207 (*Fugacium* sp.), and GTP-A6-Sy (*Symbiodinium natans*). The chemotactic activity correlated with the number of Symbiodiniaceae cells acquired by juvenile polyps. This promotes symbiosis between the polyps and Symbiodiniaceae cells [240].

## 6. Consumer Expectations in the Safety of Lectin-Based Food Products

In recent years, consumer demand for bio-active food products with functional activity are increasing, and the food industry faces newer challenges for the sustainability of commercial food products [241]. Lectins have excellent nutritional and pharmaceutical benefits, which enhance their commercial value and account for their increasing popularity in the biomedical and food industries [242,243,244]. Some food industry incorporates marine food-derived lectins in functional food or nutraceuticals to prevent calcium deficiency [241]. Although most lectins are beneficial to humans, several lectins can be of public health concern when consumed in high doses [245,246]. High doses of some lectins can cause diarrhea, vomiting, gastrointestinal damage, affect the gut microbiota, and cause nausea [247,248,249]. Thus, much attention is needed during the selection of lectins for pharmaceutical products and health care services. Furthermore, the setting of safe dosages requires scientific attention [245,246].

## 7. Conclusions

Lectins are structurally diverse molecules characterized by their unique ability to bind an array of sugar moieties and have been exploited to diagnose abnormal cells and pathogens by binding to the sugar moieties present on their surfaces through carbohydrate binding sites. The studies on the isolation, purification, and functional characterization of lectins from marine resources, primarily from fish, have augmented during the last decades. Aquatic organisms play an essential role in providing various types of lectins with multiple physiologically beneficial properties. Several clinical and sub-clinical trials have specified that marine lectins are highly likely to prevent harmful microorganisms, enhance immunomodulation, protect against cancer, and aid in disease and pathogen detection. Despite the tremendous beneficial health effects, lectin-based products may cause public health concerns, including several adverse effects in cases of improper dosages. Attention is recommended in the selection and determination of proper lectins dosages for humans. Nevertheless, marine lectins have an optimistic future of being utilized in functional foods as well as biomedical and pharmaceutical industries. 

## Figures and Tables

**Figure 2 marinedrugs-20-00430-f002:**
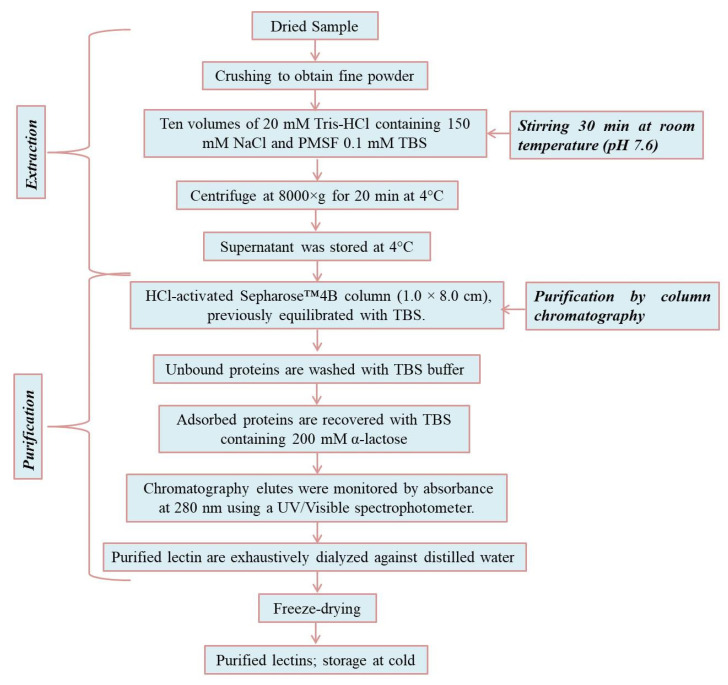
Schematic presentation of lectin extraction and purification process.

**Figure 3 marinedrugs-20-00430-f003:**
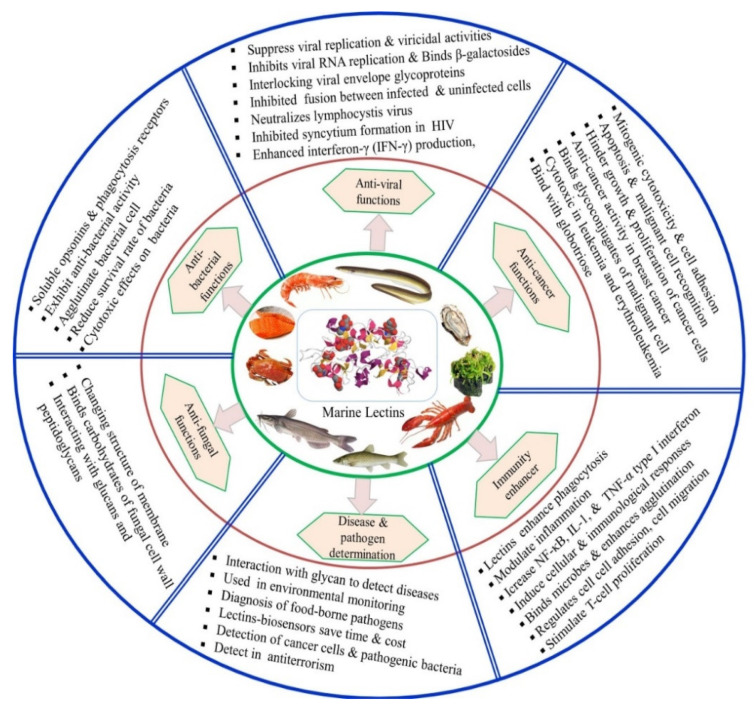
Biofunctional and immunopotential roles of marine lectin.

**Figure 4 marinedrugs-20-00430-f004:**
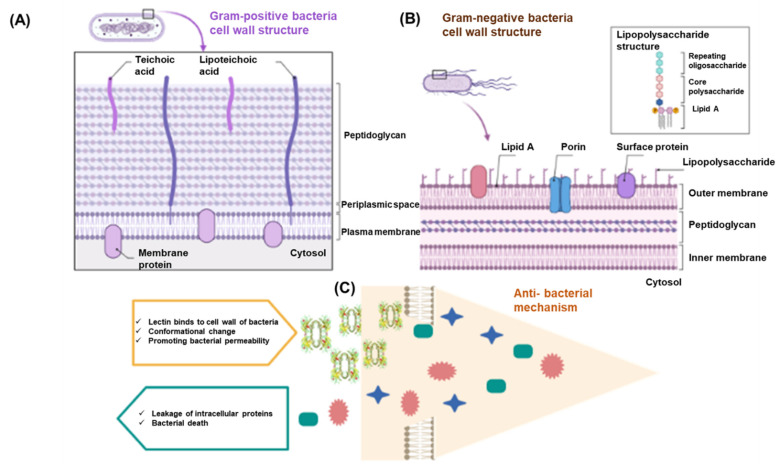
Cell wall structure of Gram-positive (**A**) and Gram-negative (**B**) bacteria. Antibacterial mechanism showing lectin binding to bacterial cells and subsequent inhibition (**C**).

**Figure 5 marinedrugs-20-00430-f005:**
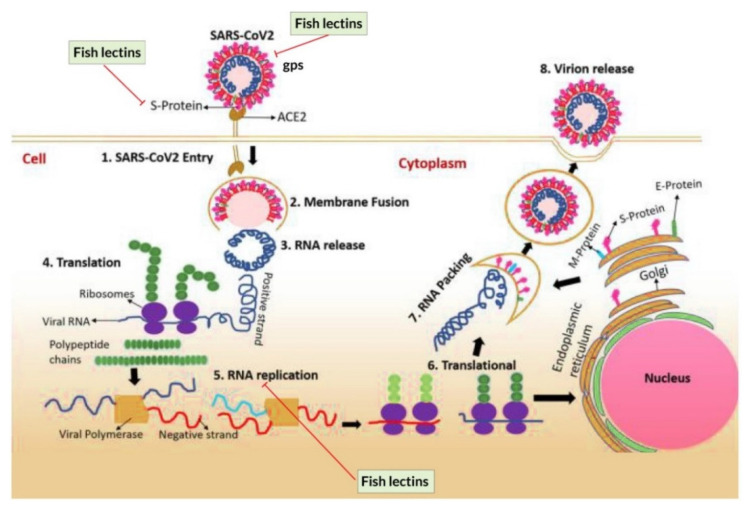
Schematic diagram of the mechanism of entry of SARS-CoV-2, viral replication, and viral RNA packing in a human cell (modified from [197]). The red line indicates the inhibition sites of SARS-CoV-2 by fish lectins. Copyright © 2020 Informa UK Limited, trading as Taylor & Francis Group.

**Figure 6 marinedrugs-20-00430-f006:**
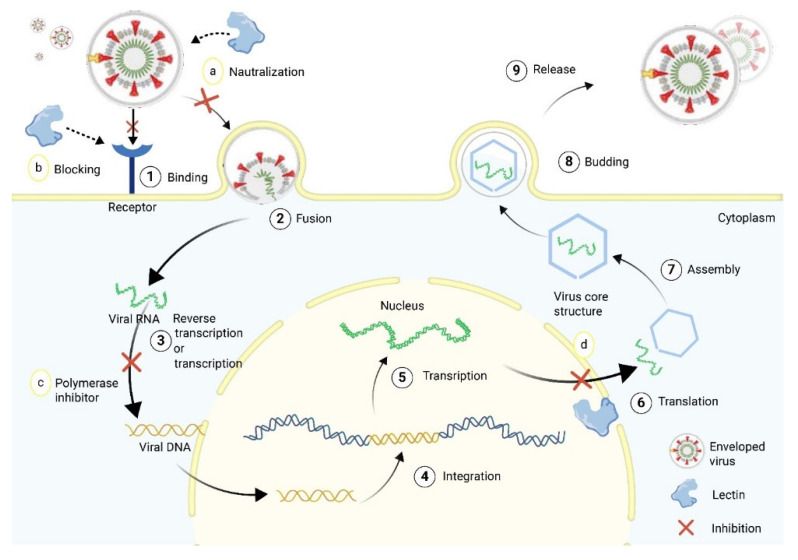
Mechanism of lectin anti-viral activity targeted different steps in the virus life cycle (redrawn from [202] with permission, lisence no. 5337420221584. Copyright © 2021 Elsevier B.V. All rights reserved.

**Figure 7 marinedrugs-20-00430-f007:**
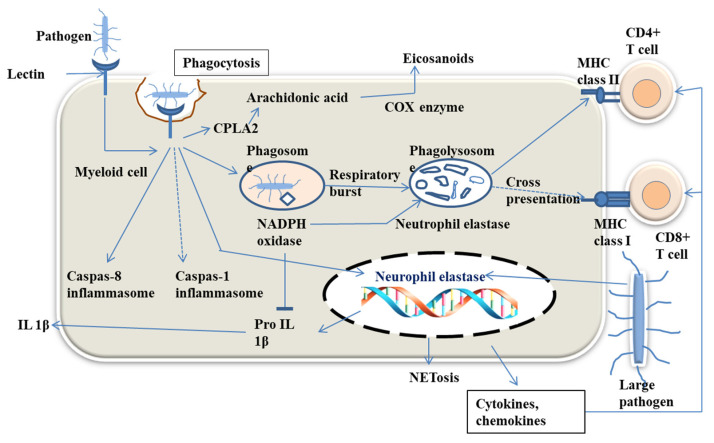
Cellular function of lectins as immunity enhancer. Abbreviations: IL-1β: Interleukin 1 beta, NADPH: Reduced nicotinamide adenine dinucleotide phosphate, CPLA2: Cytosolic Phospholipase A2, COX: Cyclooxygenase, NETosis: Neutrophil Extracellular Traps mediated necrosis, MHC: Major Histocompatibility Complex, CD4: Cluster of Differentiation 4, CD8: Cluster of Differentiation 8.

**Figure 8 marinedrugs-20-00430-f008:**
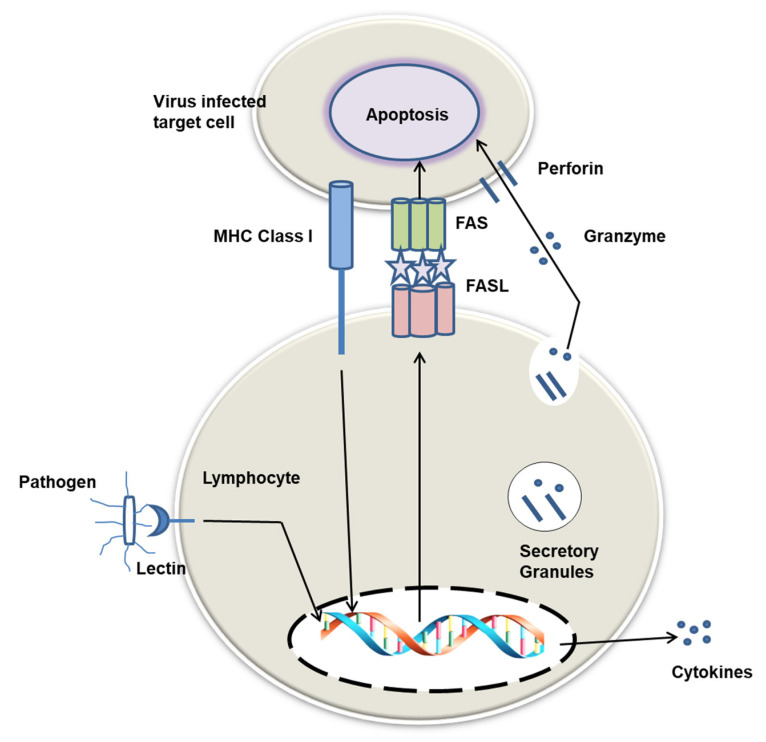
Direct recognition of pathogens and production of pro-inflammatory cytokines. Abbreviations: FAS: Fas receptor, FASL: Fas ligand.

**Table 1 marinedrugs-20-00430-t001:** Lectins extracted from different organisms and their biological functions.

Types of Organisms	Lectin Family	Specificity	Tissue Expression	Features/Functions	References
Turbot (*Scophthalmus maximus* L.)	RBL	l-rhamnose or d-galactose	Egg cortex and ovary cells	– Agglutinates Gram-positive and Gram-negative bacteria – Enhances phagocytosis	[27]
Purplish bifurcate mussel (*Mytilisepta virgata*)	RBL	Sevil, a glycan binding lectin	Mantles and gills	– Showed cytotoxic effects (apoptosis) against ovarian, breast, and colonial cancer cell line culture – Apoptosis against dog kidney cell line culture – Provides immune defense against infecting pathogens	[28]
Sea urchin (*Anthocidaris crassispina*)	RBL (SUEL-type)	d-Galactose	Eggs	– Expresses hemagglutination activity through the disulfide-linked homodimer of two subunits	[29]
Southern catfish (*Silurus meridionalis*)	RBL	l-rhamnose	Gill, barbel	– Contains two of the seven CRD (CRD3 and CRD5) described in animals – Involved in innate immunity	[30]
Nile tilapia (*Oreochromis* *niloticus*)	RBL	l-rhamnose	Liver, gills, intestines	– Contains four lectin genes with two tandem-repeated five CRD – Activates innate immune responses to infections	[31]
Catfish (*Silurus asotus*)	RBL	α-galactoside	Eggs	– Composed of 3 tandem-repeated domains which bind to globotriaosylceramide (Gb3) glycan (Gala1-4Galb1-4Glc) – Induces earlier apoptosis in Burkitt’s lymphoma cell lines – Enhances the effectiveness of anti-cancer drugs	[32,33,34]
Turbot (*Scophthalmus maximus* L.)	RBL	l-rhamnose	Egg, ovaries	– Composed two RBL with tandems repeated CRD5 of type IIIc RBL – Involved in turbot mucosal immunity	[27]
Chum salmon (*Oncorhynchus keta)*	RBL	l-rhamnose	Eggs	– Enhances intracellular Ca^++^ of Caco-2 cell monolayers	[35]
Shishamo smelt (*Osmerus lanceolatus*)	RBL	l-rhamnose	Eggs	– Exhibits two tandem-repeated domains – Binds to Raji cells through the Gb3 carbohydrate chain and induces cell death – Inhibits by melibiose	[34]
Bay scallop (*Argopecten irradians*)	CTL	Galactose	Muscle, gonad, hepatopancreas, mantle margine, and gill	– Play a crucial role in the innate immunity of bay scallop such as antimicrobial activity, non-self-recognition, and promotion nodule formation and phagocytosis – Provide strong immune response against Gram-positive and Gram-negative bacterial infection	[36]
Horsemussels (*Modiolus kurilensis*)	CTL	Glycan	Hemolymph	– Demonstrated antibacterial activities against Gram-positive and Gram-negative bacteria – Inhibits the human adenocarcinoma HeLa cells proliferation – Has immune function in *M. kurilensis* – Has pattern recognition receptors (PRR) and has interactions with Pathogen-Associated Molecular patterns (PAMPs) (e.g., mannan, PDG and LPS) – Growth inhibition of bacteria and shows agglutination activity – Potential for application in the field of biotechnology and biomedicine	[37]
Clam (*Glycymeris yessoensis*)	CTL	Peptidoglycan, LPS, β-1,3-glucan and mannan	Hemolymph	– Exhibited immune response of clam against bacterial attack – Served as PRR – Useful marker for understanding the immune system status of bivalves – Marker for studying environmental induced stress in mollusks – Synthesis of this molecule increased when animals were exposed to pathogens or environmental stresses	[38]
Kadal Shrimp (*Metapenaeus dobsoni*)	CTL	Glycan	Hemolymph	– Showed antibacterial activity against pathogenic *Aeromonas hydrophila*, *Enterococcus fecalis* and *Streptococcus iniae* challenged with Nile tilapia	[39]
Molluscan snail (*Hemifusus pugilinus*)	CTL	Mannose	Haemolymph	– Exhibited antibacterial and antibiofilm activity towards Gram-positive and Gram-negative bacteria – Showed antifungal activity against *Aspergillus niger* and *A. flavus* – Increased innate immune response	[40]
Manila clam (*Venerupis philippinarum*)	CTL	Glucan	Gill tissues and hepatopancreas	– Agglutinate Gram-positive and Gram-negative bacteria – Demonstrated prompt phagocytosis against non-self – Enhanced innate immune responses	[41]
Sea urchin (*Pseudocentrotus depressus*)	CTL	Mannose	Crushed body	– Agglutinate *E. coli* and *Lactococcus garvieae* – Act as defense molecules on the body surface of sea urchin	[42]
Sea urchin (*Pseudocentrotus depressus*)	CTL	Glycan	Tube feet	– Composed of five lectins, such as *Griffonia simplicifolia* lectin II (GSL II), wheat germ agglutinin (WGA), *Solanum tuberosum* lectin (STL), *Lycopersicon esculentum* lectin (LEL), and soybean agglutinin (SBA) – Could secret additives and provide additive power to the tube feet	[43]
Mud crab (*Scylla paramamosain*)	CTL	Glucan	Hemocytes, midgut, muscle, stomach, hapatopancreas, testis, ovaries, and heart	– Play a key role in immune-related genes and immunological parameters – Reduced the bacterial endotoxin level in vitro that renders to improve the survival rate of mud crab – Effective for mud crab aquaculture disease control	[44]
Brittle star (*Ophioplocus japonicus*)	CTL	Glucose/xylose	Whole body	– NS	[45]
Goldfish (*Carassius auratus*)	CTL	Mannose	Liver, spleen, kidney	– Calcium ion-dependent – Agglutinates rabbit erythrocytes and bacteria (*E. coli* and *A. hydrophila*, *S. aureus*) – Initiates innate immunity in the host	[46]
Turbot (*Scophthalmus maximus*) Black rockfish (*Sebastes schlegelii*) (SsLTL)	LTL	d-mannose	Skin, gill, and intestine	– Hydrophilic protein – Enhances hemagglutinating activity against fish and mice erythrocytes – Selectively binds to bacterial species including *E. tarda* and *V. anguillarum*	[47,48,49]
Giant prawn (*Macrobrachium rosenbergii*)	LTL	Mannose	Hemocytes, intestine, and hepatopancreas	– Inhibited the growth activities of microorganisms in vitro – Accelerated the bacterial clearance in vivo – Inhibited the virus replication in vivo that reduce the mortality of prawn	[50]
Sea bass (*Dicentrarchus labrax* L.)	FTL	Fucose-binding	Liver, larvae, eggs, intestine	– Embryo FBL exhibits MW of 34 kDa under reducing conditions but 30 kDa in the absence of B mercaptoethanol – Agglutinate erythrocytes	[51,52]
Striped beakfish (rock bream) (*Oplegnathus fasciatus*)	FTL	Fucose-binding	Intestines	– Modulates the expression of proteins related to viral budding and thrombin signaling (F2), which increase the viability of VHSV-infected cells	[53]
Striped Bass (*Morone saxatilis*)	FTL	Fucose-binding	Liver	– Two-tandem domains that exhibit CRS motif – Enhances innate immune responses	[54]
sea bass (*Dicentrarchus labrax*)	FTL	l-rhamnose	Liver and intestine	– Enhances phagocytosis	[55]
Steelhead trout (*Oncorhynchus mykiss*)	FTL	l-rhamnose	Eggs	– Exists in two forms: STL1 and STL2 with estimated MW of 84 and 68 kDa, respectively – STL1—noncovalently linked trimer of 31.4-kDa subunits – STL2—noncovalently linked trimer of 21.5-kDa subunits – Agglutinates rabbit erythrocytes	[56]
Sea bass (*Dicentrarchus labrax*)	FTL	l-fucose-binding	Intestine, liver	– Enhances immune defense responses in intestinal mucus and bloodstream – Upregulates gene expression and secretion of encoded proteins that are involved in both the innate and adaptive immune responses	[57]
Atlantic Salmon (*Salmo salar*)	Galectin	Glycans	Gill epithelial cell	– Two candidates: mannobiose and N-acetylgalactosamine (GalNAc) – Instant amoeba detachment – Block parasitic attachment	[58]
Striped snakehead (*Channa striatus*)	Galectin	Galactosidase	Liver	– Expression induced by epizootic ulcerative syndrome (EUS) causing pathogens such as Aphanomyces invadans – G4 peptide exhibits weak bactericidal activity against *Vibrio harveyi* – Relies on pentamer oligotryptophan (W5) at the C-terminal for its membrane disruption activity	[9]
Korean rose bitterling (*Rhodeus uyekii*)	Galectin	β-galactoside	Liver, brain, kidney, ovary, gills, spleen	– Upregulates by lipopolysaccharide – Triggers innate immunity	[59]
Turbot (*Scophthalmus maximus* L.)	Galectin	β-galactoside	Skin and brain	– Strong binding potentials to microbial ligands – Enhances immune response against infection	[60]
Euryhaline rotifers (*Brachionus* *Plicatilis* and *Proales similis*)	Galectins	Carbohydrate-binding domains with long N-terminal region (i.e., ~100 amino acids) β-galactosyl binding lectins		– C-type lectins- – Regulate innate immunity by enhancing microbial opsonization and melanization through prophenoloxidase enzyme activation – Activation of complement system – Serve as mate recognition pheromone – Galectins- – Assist cell adhesion, – Maintain cellular homeostasis – Help self/non-self and microbial recognition	[61]
Sea Hare (*Aplysia kurodai*)	GBL	d-galacturonic acid and d-galactose	Eggs	– Showed a moderate toxicity to *Artemia* nauplii – Apoptosis to cell death – Worked against the growth of erythroleukemia cells of human – Antifungal and antibacterial activities and involved in the defense of sea hare embryo – Suppressed the growth of the tumors, such as Ehrlich ascites carcinoma	[62]
Pikeperch (*Sander lucioperca*), rainbow trout (*Oncorhynchus mykiss*), and maraena whitefish (*Coregonus maraena*)	Siglecs (Siglec1, Siglec15, CD22, and myelin-associated glycoprotein (MAG))	Sialic-acid-binding	Head kidney, liver, gills, spleen, heart, and muscle	– MAG contains immunoreceptor tyrosine-based inhibitory motif (ITIM) – Some pathogens can express sialic acids which is identified by Siglecs – Fish associated with 4 types: Siglec1, CD22, myelin-associated glycoprotein (MAG) and Siglec 15 – Influences the cellular reactivity against damage-associated molecular patterns (DAMPs)	[63]
Marine sponge (*Aplysina fulva*)	AFL	Galactose	Crude extract	– Marine sponge showed interesting bioactivities such as antitumor, antiviral, and antibacterial activity – Mitogenicity, modulatory, and cytotoxicity activity on mammalian glutamate-gated ion channels – Reduce the biomass biofilm of the *E. coli*, *Staphylococcus aureus*, and *S. epidermidis*	[64]

Galactose binding lectin (GBL), rhamnose binding lectin (RBL), B-type Lectins (BTL), Lily type lectin (LTL), C-type lectin (CTL), F-type lectin (FTL), viral haemorrhagic septicaemia virus (BTL), mucin-binding lectin (AFL), molecular weight (MW), carbohydrate recognition domain (CRD), carbohydrate recognition sequence (CRS), viral hemorrhagic septicemia virus (VHSV), immunoglobulin-like lectin (IgTL), not reported (NR), not studied (NS).

## Data Availability

Not applicable.

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
