# Peer review of "An Update of Lectins from Marine Organisms: Characterization, Extraction Methodology, and Potential Biofunctional Applications"

_marinedrugs, 2022, doi:10.3390/md20070430_

Round 1

Reviewer 1 Report

The manuscript entitled "An update of marine and freshwater lectins: characterization, extraction methodology, and potential biofunctional applications" is a review of marine, primarily fish-derived lectins. It provides information on their characterization, extraction and functions. Overall this could be a good review and I was looking forward to reading it. Unfortunately,  and the manuscript does need significant revision prior to publication.

The review is too long and could be significantly improved by more focus. Some sections, such as the section on lectin purification, should be omitted. Tables such as Table 2 need significant revision as they evidence a misunderstanding of the cited literature. The organization of the functional section should be altered to lead with antibacterial uses as this is the longest and best-referenced section [it also need figures as there are none for this entire section]. Some short functional sections should be omitted completely as they do not adequately cover the area they describe and they do not add significantly to the review. I would suggest omitting sections 5.7-5.11; most of these are only one or two sentences long. 

That being said, with revision should be worth re-submission to  Marine Drugs.  

Other specific recommendations:

1. The authors should use the Oxford dictionary definition rather than the Webster's dictionary definition of the term "lectin". To state that lectins are "non immune" carbohydrate-binding proteins only to then describe numerous instance of immune function for lectins is incongruent. The Oxford definitions does not include a statement on "non immune" function. 

2. The language in individual sections is often redundant. lets are frequently reported to have "unique sequence motifs" or "structural folds" or both. Also, the term "potent" is used indiscriminately throughout the manuscript to describe widely variant levels of activity. Finally, the authors veer between ug/mL and molar units in describing biological activities. These should all be in the same units to better allow the reader to compare potencies.

3. Section 2 would be improved by actually showing some of the lectin structures being described in the text. There are currently none and there are numerous structures available in PDB. 

4. The extraction and purification section should be omitted (including Figure 1). the procedures described are only a snapshot of the methods published for the purification of lectins and serve minimal benefit to the reader. Describing column gradients for isolated procedures is not comprehensive. This section detracts from the overall review. 

5. Section 4 is inadequate. To have only one paragraph on the physiological functions of lectins in their host organisms shows the authors did not sufficiently research this area. Certainly not in sufficient depth for a review which is intended to guide the reader to the relevant primary literature. This section should be significantly improved prior to acceptance. 

6. Section 5.1 discusses antiviral activity. The authors start by evidencing a misunderstanding of lectin MOA against viruses. Lectins do not "inhibit viral replication" they inhibit viral entry. Lectins are not virucidal unless they can be shown to actively destroy a virus. Viruses are not "giving unless in a host cell. Lectins are extracellular active agents. As such, they can "inactivate" a viral particle (make it so it cannot enter a cell by binding to surface glycoproteins) but there are very few if any cases of "virucidal" lectins. Throughput this section and in Table 2 the authors mis-understand or mis-represent the primary literature. For example, griffith sin doe not show "enhanced binding" to gp12o in the present to galactose, xylem etc., those monosaccharides just do not reduce its binding ( as wit mannose and glucose). In Table 2, the authors quote the amount and animal model used to produce anti-Griffithsin antibodies rather than any of the antiviral studies undertaken with this lectin. Significant effort should be expended by the authors to improved this section and Table 2 to adequately reflect the primary literature. Table 2 should also include more relevant columns than "Duration"  and "Test types" . It should have antiviral activity, virus (including strain) and all activity should be standardized. It should all include all relevant antiviral or antibacterial literature not just a random example for certain lectins. The table is far from comprehensive.

Finally, the last two paragraphs of this section are not well written. The final tow paragraphs starting on line 433 talk about small molecules from marine organisms (alkaloids, terpenoids, polysaccharides, peptides etc.) more than they do about lectins. 

7. The authors spend too much text on background for SARS-coV-2 for only a paragraph on potential lectins with anti-coronavirus activity. The readership of Marine Drugs is largely aware of all this background information on the virus. It does not need to be repeated in this review (as it was not done for any other virus). 

Reviewer 2 Report

I have gone through the research manuscript entitled: “An update of marine and freshwater lectins: characterization, extraction methodology, and potential biofunctional applications. This manuscript has indicated the many aquatic lectins biochemical properties and it will contribute to provide the useful information for researcher focusing on marine lectins. To improve the manuscript, reviewer described some finding as below. The language of the manuscript is written well and expressed clearly.  

Some corrections are suggested as mentioned below:

In the section of “Fish lectins”, many of the structural and functional properties were described such as “Rhamnose binding lectins”, “Fucose binding lectins” and “C-type lectins” and so on. In general, classification of lectins are dependent on their primary structure rather than monosaccharide binding property as Galectin, C type lectin which are known as representative lectin family. Rhanmose binding lectins can also categorize as SUEL type lectin (Tateno, Biosci Biotechnol Biochem., 2010) which were firstly discovered the primary structure of sea urchin unfertilized egg lectin (Ozeki et al., Biochemistry, 1991). In addition, this article title contains the word of “update. Due to this context, adding the new information of lectins containing novel primary structure will be improve the manuscript quality (Rajan et al., Fish Shellfish Immunol. 2017; this is just one of the example). In addition, you explain the physiological function of Siglecs from fishes on another section but no indication in the table 1. It is better to summarize in table. To update the primary structure of aquatic lectins, novel lectins having unique primary structure should be included.

In the table 1, specificity of lectin was shown to bind the monosaccharides or disaccharides. In these days, many of glycobiological techniques are improved by many glycobiologists, for instance glycan array and frontal affinity chromatography. So the details of glycan binding profile could display also such as you already indicated cat fish (Silurus asotus) egg lectin is Gb3 binding lectin. Other lectins should also indicate more details of glycan binding profiles rather than showing the monosaccharide or disaccharides.

In page3, line 135, 139, Globotriaosylceramide is indicated as “GB3”. But the table shows “Gb3”. It should integrate as “Gb3”.

Figure 3 is summary of the mechanisms of SARS-CoV-2 entry to human cells. It is better to show the scheme that how the fish lectins exert to inhibit the infection of SARS-CoV-2 virus instead of showing the general mechanism.

In the section of “5.6. Lectins act as a biosensor for disease and pathogen determination”, there was no indication of aquatic lectins are involved. It is better to omit this section or should indicate that aquatic lectin is useful for biosensor and show the example for this.

Round 2

Reviewer 1 Report

The changes are acceptable. 

Author Response

We thank Reviewer for time and essential comments, which helped to improve the overall quality of the manuscript.